# Quantifying the impact of a broadly protective sarbecovirus vaccine in a future SARS-X pandemic

COVID-19 has underscored the need for more timely access to vaccines during future pandemics. This has motivated development of broad-spectrum vaccines providing protection against entire viral families, which could be stockpiled and deployed rapidly following detection. Using mathematical modelling, we assess the utility of a broadly protective sarbecovirus vaccine during a hypothetical SARS-X outbreak, for a range of implementation strategies including ring-vaccination, spatial-targeting and mass vaccination of high-risk groups. Broadly protective sarbecovirus vaccine ring- or spatial strategies alone are insufficient to contain epidemics driven by a SARS-CoV-2-like virus, but when paired with rapid isolation and quarantine, can achieve containment of a SARS-CoV-1-like virus. Where suppression fails, broadly protective sarbecovirus vaccine utilisation still reduces the effective reproduction number and slows epidemic growth - buying valuable time for health-system response and virus-specific vaccine development. Vaccination of high-risk populations with the broadly protective sarbecovirus vaccine ahead of virus-specific vaccine availability could reduce mortality and enable shorter and less stringent non-pharmaceutical interventions to be imposed; results are sensitive to vaccine properties (e.g., efficacy), health system capabilities (e.g. rollout speed) and timeline to virus-specific vaccine availability. Our modelling suggests that broadly protective sarbecovirus vaccine delivery to those aged 60+ years could have averted 21-78 % of COVID-19 deaths during the pandemic's first year, depending on the size of the stockpile. Realising this potential impact will require investment in manufacturing, delivery capacity and equitable access ahead of future pandemics.

COVID-19 highlighted the crucial role of vaccination in reducing disease burden and mitigating socio-economic impact during pandemics. An estimated 14–20 million deaths were averted due to vaccinations in their first year of use[1] and enabled lifting of societal restrictions that carried significant socio-economic costs[2]. Development and authorisation of highly efficacious vaccines against COVID-19 within a year represents a significant achievement, given that development pipelines typically take 10 or more years[3]. Despite this accelerated timeline,

>1.5 million confirmed COVID-19 deaths occurred during this time-period[4]. Moreover, access to doses was highly unequal between the global north and global south[5]. This is despite the additional lives that equitable allocation strategies could have saved[6–9].

SARS-CoV-2 is unlikely to be the last pandemic faced by the world[10]. The frequency of pathogen spillover and intensity of consequent epidemics are projected to increase in the future[11,12]. This has motivated interest in reducing vaccine development timelines,

✉ e-mail: cwhittaker@berkeley.edu; a.ghani@imperial.ac.uk

including recent initiatives aiming to enable development, authorisation and manufacture of vaccines against a novel pathogen within 100 days of identification[13–15]. However, other work has indicated that existing approaches to vaccine development are unlikely to achieve development timelines of <250 days[16]. A further limitation of these approaches is their reactive nature: pathogen-specific vaccine development depends on detecting and sequencing the novel pathogen's genome. This limits the timeliness of strategies aiming to develop vaccines in response to the emergence of a novel pathogen; such delays lead to substantial human mortality and/or the necessity of significant (and costly) control measures in the form of non-pharmaceutical interventions (NPIs).

Research has therefore focused on alternative approaches to vaccine development that might facilitate more rapid availability. Broad-spectrum vaccines providing protection against multiple viruses in the same family or sub-family could be manufactured and stockpiled ahead of a pandemic, ready for rapid deployment following identification of a novel pathogen outbreak. Several vaccines aimed at providing broad and robust protection to a range of coronaviruses are under development[17,18]. Many have demonstrated an ability to induce broad neutralising antibodies in mice; several have demonstrated this in non-human primates. Potent pan-sarbecovirus neutralising antibodies have been identified in humans previously infected by a range of coronaviruses[19–21], suggesting that eliciting broad-spectrum protection should be possible. These candidates span a range of approaches, including mosaic nanoparticles containing spike receptor binding domains from multiple sarbecoviruses[22]; chimeric spike mRNA vaccines[23]; and antigens based on epitopes conserved across multiple coronaviruses[24].

Previous modelling studies focused on the US have highlighted that a stockpiled pan-coronavirus vaccine could avert significant disease burden and be cost-saving, even in the context of modest vaccine efficacy and partial rollout[25]. Complementary work evaluating hypothetical universal influenza A vaccines reaches similar conclusions, underscoring the importance of breadth, durability and speed of deployment[26]. Here, we use a mathematical modelling framework to evaluate the utility of a broadly protective sarbecovirus vaccine (BPSV) during a hypothetical global future sarbecovirus ("SARS-X") pandemic. We explore a wide range of implementation strategies, including ring- and spatial-vaccination[27–29] for outbreak suppression, and mass vaccination of vulnerable populations for disease burden mitigation, across a diverse range of settings. Our work highlights substantial potential public-health impact from widespread and rapid access to a BPSV during a SARS-X pandemic, and the potential utility of broad-spectrum vaccines as tools to support future pandemic preparedness and response efforts.

## Results

### BPSV ring-vaccination could support outbreak containment efforts of a sarbecovirus similar to SARS-CoV-1 but not SARS-CoV-2

We developed a stochastic branching-process framework to explore the potential for a BPSV to support containment of a hypothetical SARS-X outbreak via a ring vaccination approach[28] (Fig. 1A). We considered two "archetype" sarbecoviruses—one similar to SARS-CoV-1 (Fig. 1B, mean generation time 12 days, 0% presymptomatic transmission, 0% asymptomatic infections) and one similar to SARS-CoV-2 (Fig. 1C, mean generation time 6.75 days, 35% presymptomatic transmission, 15% asymptomatic infections). We assumed ring-vaccination enabled 80% of symptomatic contacts to be vaccinated[30] and explicitly model quarantine and isolation, which can occur either through contact tracing (in which 47% of contacts identified after their infector becomes symptomatic enter quarantine a mean of 4 days after exposure) and symptom-based self-recognition (in which 90% of individuals isolate a mean of 4 days after their own symptom onset). In both cases, quarantine/isolation is assumed to cut onward transmissibility by 65%[31]. We further assumed a BPSV efficacy of 35% against infection, that

breakthrough infections in vaccinated individuals have a 35% reduced infectiousness, and a delay of 2 days between the identification of a symptomatic index case and their contacts receiving the vaccine. We also carried out detailed sensitivity analyses varying assumptions around the BPSV properties, pathogen parameters and quarantine efficacy (see Supplementary Table 1 and below).

Across both pathogen archetypes, the proportion of outbreaks contained decreased as R0 increased and decreased with the delay between vaccination and protection (vaccine delay to protection, VDP). For the "SARS-CoV-1-like-virus" and contexts without quarantine, administering vaccines with a VDP of ≤1 week contained all outbreaks for R0 ≤ 1.5 (Fig. 1B, bottom panel) and successfully reduced transmission such that the effective reproduction number ($R_{Eff}$) was below 1 (Fig. 1B, top panel). However, a vaccine with a VDP of 2 weeks resulted in minimal impact, with less than 1% of outbreaks contained across all values of R0 (Fig. 1B). The addition of contact quarantining improved control prospects overall but the BPSV still failed to provide additional benefit at longer VDP values (Fig. 1D). For the "SARS-CoV-2-like-virus", with high pre-symptomatic transmission and a higher asymptomatic fraction, containment was only possible with a VDP ≤ 2 days and for epidemics with R0 ≤ 1.5 without quarantine (Fig. 1C), and VDP ≤ 2 days and R0 ≤ 2 when quarantine of symptomatic contacts was included (Fig. 1E). Addition of the BPSV did offer benefit in some scenarios where "control" was incomplete and $R_{Eff}$ remained above 1. In these contexts, deployment of the BPSV resulted in a reduction in $R_{Eff}$, attenuating transmission and thereby slowing epidemic expansion relative to scenarios in which the BPSV was unavailable (Fig. S1).

Prospects for control are sensitive to vaccine characteristics and pathogen properties (Fig. 1F–H). When the VDP equalled or exceeded the mean generation time (Tg/VDP = 1, Fig. 1D), BPSV ring-vaccination did not contain the outbreak (Fig. 1F). Containment prospects were improved for relatively longer generation times (Tg/VDP ≥ 3) and in contexts where a more efficacious vaccine (75% efficacy) was assumed (Fig. S2A). In general, even when containment was incomplete (i.e., some epidemics were not successfully suppressed), the addition of a BPSV led to reductions in R and slowed epidemic progression (Fig. S2B, C). Increasing vaccine efficacy against infection led to an increasing fraction of successfully contained outbreaks for R0 ≤ 2 but failed for higher R0 except when combined with efficacious isolation and quarantine of contacts (Fig. 1G and Fig. S3). For viruses with higher levels of presymptomatic transmission, containment was generally less likely (Fig. 1H and Fig. S4).

### Spatially-targeted vaccination strategies using a BPSV must be accompanied by highly sensitive surveillance systems to be impactful

We explored the impact of spatially-targeted vaccination strategies utilising the BPSV for containment, similar to the approaches that have recently been utilised during Ebola outbreaks[29]. We assumed that BPSV vaccination would be triggered by the detection of a cluster of hospitalised cases. Following detection, and a 2-day operational delay, all individuals within a given spatial radius of the home address of the hospitalised case(s) would be vaccinated (Fig. 2A). We used the same pathogen archetypes described above, where 95% of "SARS-CoV-1-like-virus" are infections hospitalised and 5% "SARS-CoV-2-like-virus" infections are hospitalised and assumed a VPD of 7 days (i.e., a 7-day delay between receiving a vaccination and protection developing).

For "SARS-CoV-1-like-virus" with R0 < 2, spatially-targeted vaccination could contain outbreaks across all surveillance system sensitivities considered in contexts without quarantine (Fig. 2B), and R0 < 2.5 in contexts when the BPSV is combined with contact quarantining (Fig. 2D). However, for "SARS-CoV-2-like-virus" (Fig. 2C), containment with spatially-targeted vaccination with R0 < 2 occurred only in highly sensitive surveillance scenarios. The addition of contact quarantine improved containment prospects and enabled a greater

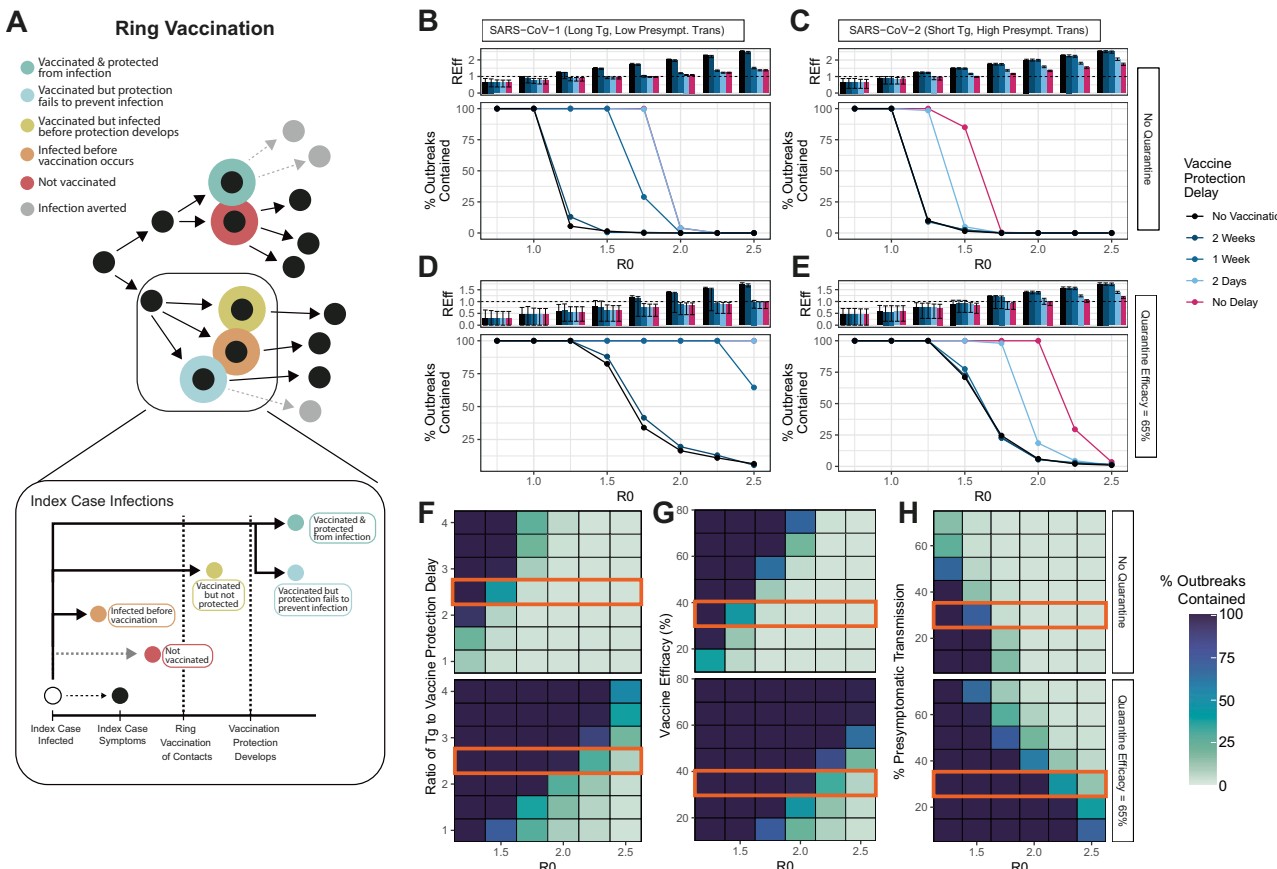

**Fig. 1 | Exploring the prospects for outbreak containment via ring vaccination strategies using a broadly protective sarbecovirus vaccine (BPSV).** A stochastic branching-process-based approach was used to explore the impact of a BPSV on outbreak containment efforts utilising ring or spatially targeted vaccination strategies, and the factors most critical to control. **A** Schematic illustrating the ring-vaccination framework. **B** The effective reproduction number (Reff, top panel) and % of outbreaks controlled (bottom panel) via BPSV ring-vaccination and its dependence on R0 (x axis), for a "SARS-CoV-1-Like" virus and where no additional control measures are implemented. Black indicates scenario without BPSV, coloured lines/bars indicate different assumptions around vaccine delay to protection (VDP). For the Reff plot, bars indicate the average Reff across 100 stochastic simulations, and the error bars indicate the 95% confidence interval of the mean of those simulations. **C** As for **B** but for a "SARS-CoV-2-like" virus. For % outbreaks controlled plot, this is the percentage of 100 stochastic simulations that are successfully controlled. **D** As for **B**, but assuming that some infected symptomatic individuals quarantine and isolate such that onward transmission is reduced by 65% following isolation. **E** As for **C** but with the addition of quarantine. **F** Sensitivity analysis exploring how the % of outbreaks contained varies with R0 (x axis) and the ratio of the generation time to the VDP (Tg/VDP), in situations without quarantine (top heatmap) and with quarantine (bottom heatmap). Orange rectangle indicates the value held constant for other sensitivity analyses. **G** As for **F** but for vaccine efficacy against infection. **H** As for **F**, but % of presymptomatic transmission.

---

fraction of outbreaks to be controlled in contexts where surveillance is less sensitive (Fig. 2E), and in instances of incomplete outbreak containment, the addition of the BPSV slowed epidemic progression relative to scenarios where it was absent. This was particularly the case where both R0 and surveillance system sensitivity were high (Fig. S5). We undertook sensitivity analyses examining how the proportion of simulated outbreaks contained varied as a function of: i) R0; ii) the ratio of the vaccination campaign's radius to the average distance separating infections (Fig. 2F and Fig. S6); iii) BPSV efficacy (Fig. 2G and Fig. S7); and iv) the number of hospitalisations required to trigger the vaccination campaign (Fig. 2H and Fig. S8). Increased vaccination campaign radius, improved vaccine efficacy and increased surveillance sensitivity were all associated with an increased fraction of outbreaks being contained. However, none of the scenarios contained outbreaks with R0 > 2 when quarantine of contacts was absent. The addition of quarantining improved containment prospects

### Vaccination of high-risk populations with a BPSV during a SARS-X outbreak could significantly reduce mortality and limit need for NPIs

We next adapted a dynamical model of SARS-CoV-2 transmission[1] to explore the utility of a stockpiled BPSV in providing rapid protection of

high-risk groups (here, assumed to be those aged 60+) to reduce disease burden during a hypothetical SARS-X outbreak in a population with a demography matching the median age-distribution of the World Bank's upper middle-income countries (UMICs). In our simulations, pathogen spillover is followed by undetected circulation in the community, with resulting hospitalisations leading to pathogen detection and identification. Virus-specific vaccine (VSV) development is triggered at this time, and takes either 250 or 100 days depending on the scenario considered (representing "realistic" and "ambitious" vaccine development timelines[13]). After an assumed delay of 7 days (to allow activation of stockpiles), mass vaccination of the at-risk population with the BPSV begins. Vaccination switches to the VSV once it becomes available, which is distributed to those aged 60+ first (including those who received the BPSV, who are boosted with the new vaccine) and then to the rest of the population. For both vaccines, we assume administration is via a single-dose regimen.

To explore the benefits of a BPSV, we compared a scenario where both the VSV and BPSV are available to a baseline scenario in which only the VSV is available. In both cases, the VSV is available only after the delay associated with its development, whereas the BPSV is available far sooner. We assume the BPSV has 75% efficacy against severe disease and 35% efficacy against infection, whilst the future VSV has

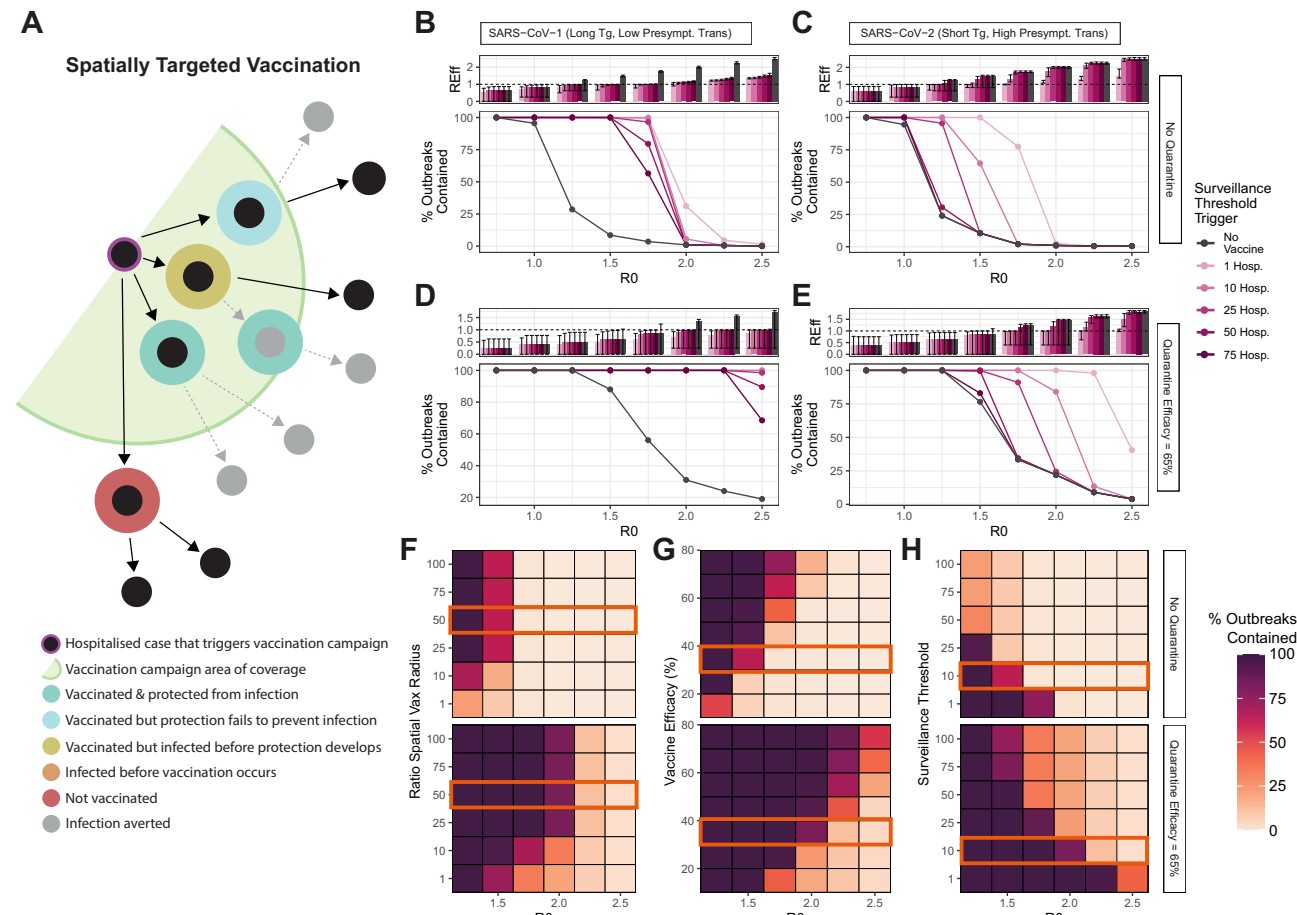

**Fig. 2 | Exploring the prospects for outbreak containment via spatially-targeted vaccination strategies using a broadly protective sarbecovirus vaccine (BPSV). A** Schematic illustrating the spatially targeted vaccination framework. **B** The effective reproduction number (Reff, top panel) and % of outbreaks controlled (bottom panel) via BPSV ring-vaccination and its dependence on R0, for a "SARS-CoV-1-Like" virus and where no additional control measures are implemented. Black indicates scenario without BPSV, coloured lines/bars indicate different assumptions around the number of hospitalisations required to trigger the spatially-targeted vaccination campaign (surveillance threshold), all assuming a 7 day VPD. For $R_{Eff}$ plot, bars indicate the average Reff across 100 stochastic simulations, and the error bars the 95% confidence interval of the mean of those simulations. **C** As for (**B**), but for a "SARS-CoV-2-like" virus. For % outbreaks controlled plot, this is the percentage of 100 stochastic simulations that are successfully controlled. **D** As for (**B**), but assuming that some infected symptomatic individuals quarantine and isolate, which reduces onward transmission by 65%. **E** As for (**C**), but with the addition of quarantine. **F** Sensitivity analysis exploring how the % of outbreaks contained varies with R0 (x axis) and the ratio of the vaccination campaign spatial radius to the average distance between infections, in situations without quarantine (top heatmap) and with quarantine (bottom heatmap). Orange rectangle indicates the value held constant for other sensitivity analyses. **G** As for **F** but for vaccine efficacy against infection. **H** As for **F** but for the surveillance threshold.

---

vaccine efficacy of 95% against severe disease and 55% against infection. We assume a R0 of 2.5, in-keeping with initial estimates of ancestral SARS-CoV-2 in Wuhan[32] (see Fig. S9 for results with different R0 values). We illustrate impact with vaccination rates of 3.5% of the country's population per week, such that the 60+ age-group is vaccinated within 4 weeks (Fig. 3A)—this corresponds to 1.55 million doses per week in a population of 40 million in the representative demography selected (where 15% of the population are aged 60+), and is derived from vaccination rates observed in UMICs during the COVID-19 pandemic[33]. We considered a range of scenarios reflecting differences in the stringency, duration and triggers for NPIs (Fig. 3B) with NPI days ranging from 0 (no intervention) to 80 for VSV development of 100 days, and 0–175 for VSV development of 250 days.

The BPSV has the greatest impact if no NPIs are implemented. When the VSV is available after 250 days, projected deaths per 1000 population were reduced from 8.6 without the BPSV to 3.85 with the BPSV. Similar results are obtained when the VSV is available after 100 days (Fig. 3C, NPI Scenario 1). With more stringent NPIs, the BPSV averts fewer deaths, but its impact depends on the time taken for the VSV to become available - if the VSV is developed and deployed more

rapidly, the relative benefit of the BPSV is reduced. For example, with a short period of stringent NPIs during the BPSV vaccination campaign followed by a minimal set of NPIs afterwards (Fig. 3C, NPI Scenario 6), the BPSV reduces deaths from 7.6 to 3.6 per 1000 population, a 53% reduction if the VSV is available after 250 days. Impact is more limited if the VSV is available after 100 days. More generally, for the 250-day development timeline, availability of the BPSV limits mortality to levels below all but the most stringent NPI scenarios when the BPSV is absent (Fig. 3C, NPI Scenario 9), enabling shorter and less stringent NPIs to be in-place for the same total disease burden. BPSV impact was high in R0 = 3.5 scenarios across all VSV development timelines considered. In R0 = 1.5 scenarios, BPSV impact was minimal except for the longest VSV development timeline (365 days) (Fig. S9).

BPSV availability also affects the level of NPIs required to limit disease burden to the level observed when only the VSV is available (Fig. 3D). For a 250-day development timeline and NPI Scenario 6 (short period of stringent NPIs during BPSV vaccination campaign followed by minimal NPIs), BPSV availability reduced the number of NPI days required from 136 to 51 (63% reduction) (Fig. 3D, lower panel).

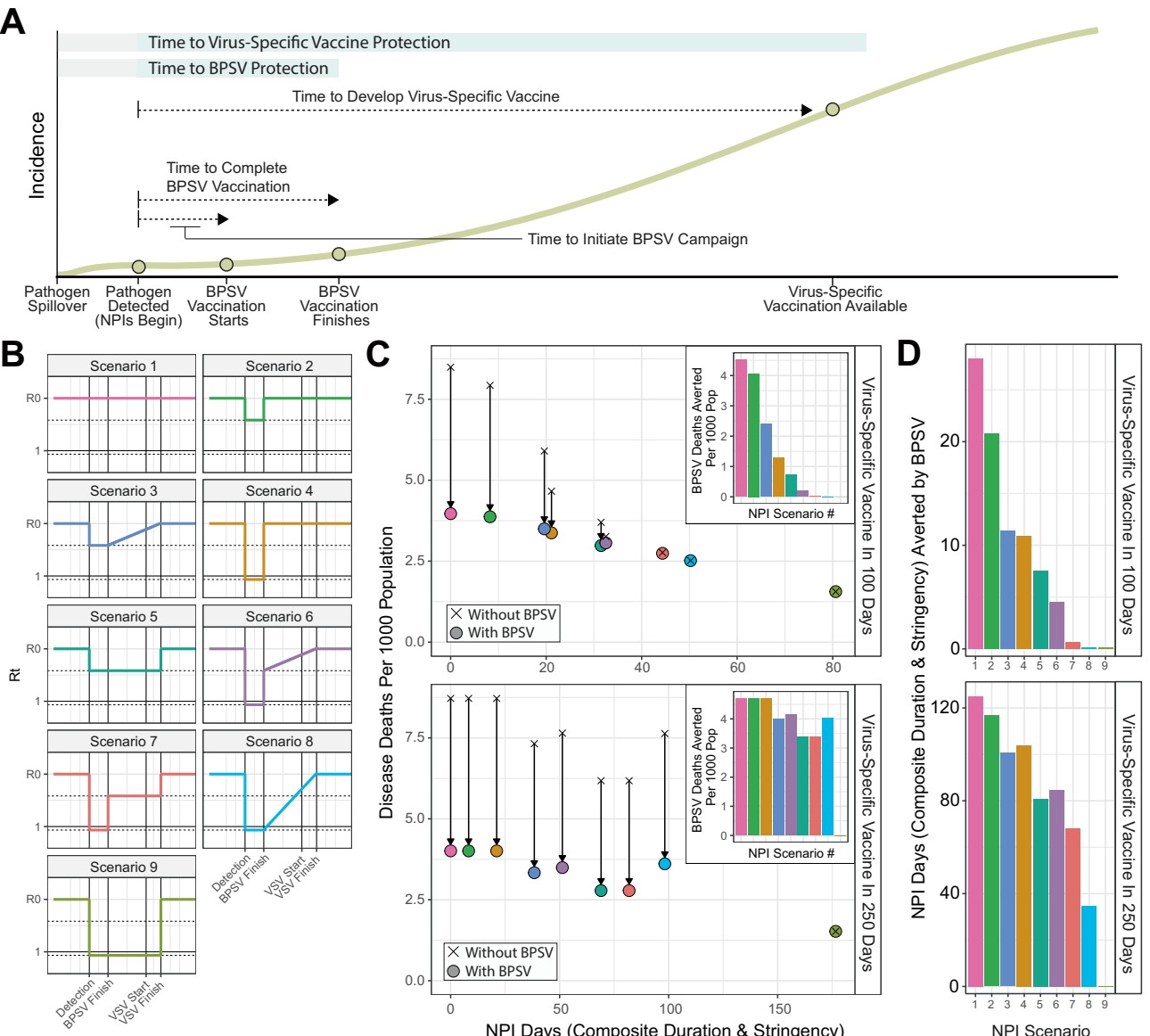

**Fig. 3 | The potential impact of BPSV mass-vaccination campaigns on disease burden during a future SARS-X pandemic.** Dynamical modelling of BPSV mass-vaccination of priority groups (those aged 60+) following pathogen detection during a hypothetical SARS-X pandemic. **A** Illustrative figure of simulated scenarios and the timing of key events. **B** Time-varying reproduction number (Rt) profiles for the different non-pharmaceutical intervention (NPI) scenarios imposed in response to the epidemic that are considered for the analyses presented here. These Scenarios differ by assumed stringency (either no measures, a minimal mandate reducing transmission by 25% or stringent measures reducing Rt to 0.9), duration (either until the BPSV campaign is completed or the disease-specific vaccination campaign is completed) and the nature by which these NPIs are relaxed (either instantaneous or gradual). **C** BPSV impact on disease burden for each NPI scenario, assuming the VSV is available 100 days (top-panel) or 250 days (bottom-panel) following detection, for an R0 of 2.5. Uncoloured crosses indicate scenarios without BPSV (VSV only); points indicate scenarios where BPSV is available, coloured according to NPI scenario. Inset panels show deaths averted by the BPSV, coloured by NPI scenario. **D** BPSV impact on the need for NPIs for the same disease burden. For each NPI Scenario, the Pareto frontier was constructed for the VSV-only scenario, and used to calculate how many fewer NPI days can be imposed in the BPSV scenario whilst still limiting disease burden to the level observed in the corresponding VSV-only scenario.

For a 100-day development timeline and the same NPI Scenario, BPSV availability reduced the NPI days by only 4 days (14%) (Fig. 3D, upper panel). In general, longer VSV development timelines necessitate more protracted and stringent NPIs to limit accrued disease burden to the same level. This is where BPSV availability reduces the most NPI need.

### BPSV availability could have substantially reduced mortality during the COVID-19 pandemic

Given the impact that NPIs have on assessing the value of the BPSV, we explored the potential impact that a stockpiled BPSV could have had on COVID-19 mortality in the pandemic's first year (Fig. 4A), using published model fits calibrated to excess mortality data[34]. We assume BPSV vaccination begins following 1000 globally reported COVID-19 deaths (see Fig. S10 for sensitivity analyses) and with rates of vaccination specific to World Bank income groups (derived from Our World In Data[33]). We also varied assumptions about the size of the BPSV stockpile (and hence fraction of eligible population able to be vaccinated)—this ranged from 40% ("Low" coverage) to 80% ("High" coverage) and also explored a scenario in which the size of the stockpile maintained varied by country ("Variable" coverage) in a manner determined by its World Bank Income Group.

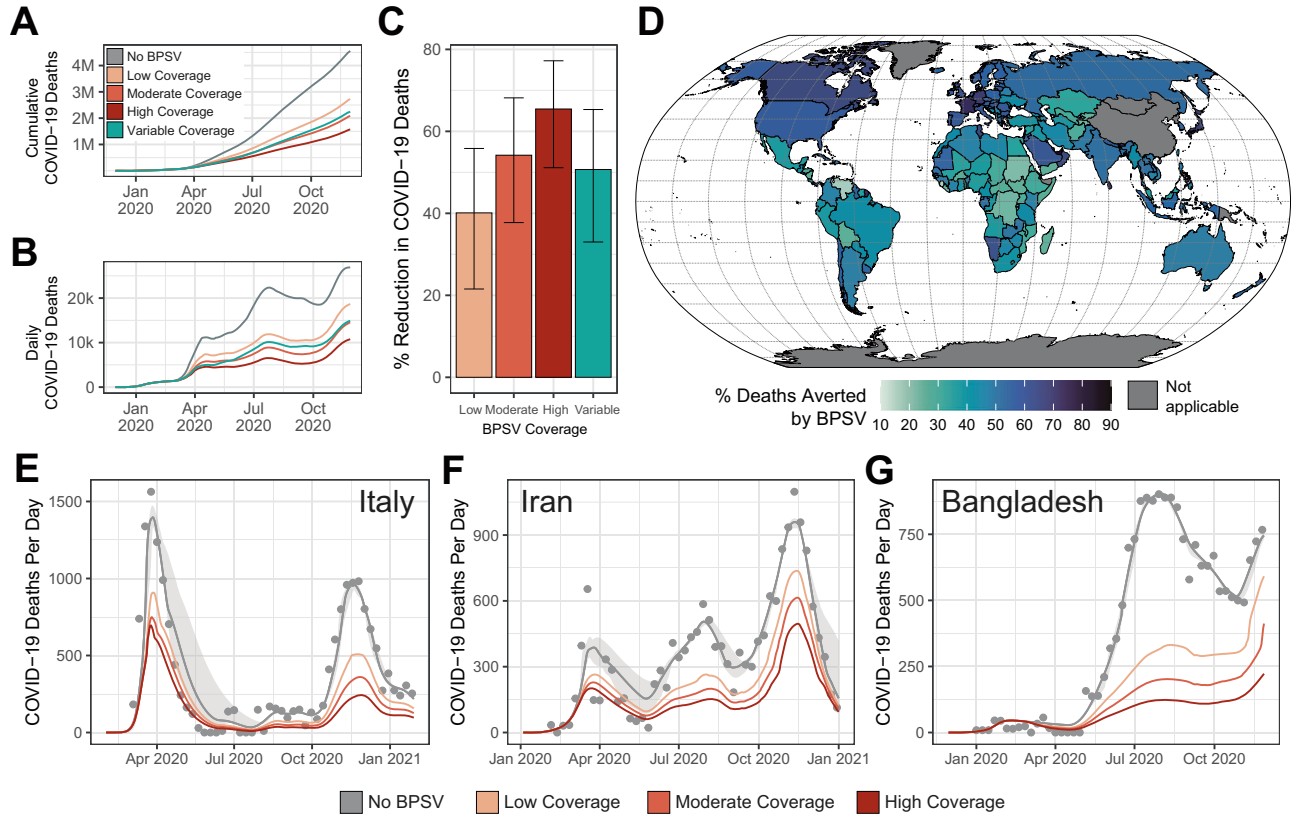

**Fig. 4 | Retrospective evaluation of BPSV impact during the COVID-19 pandemic in selected countries.** Analyses using published model fits calibrated to excess mortality data[34] retrospectively assessed the potential impact of BPSV availability on COVID-19 mortality during the SARS-CoV-2 pandemic, under various assumptions about the size a BPSV stockpile countries have access to. **A** Cumulative global COVID-19 deaths during the first year of the pandemic without (grey) the BPSV, and with the BPSV (coloured lines). Low coverage = BPSV stockpile size sufficient to vaccinate 40% of elderly population; Moderate coverage = 60%; High coverage = 80%. Variable coverage indicates size of stockpile varies according to the World Bank Income Group each country belongs to (LIC = 20%, LMIC = 40%, UMIC = 60%, HIC = 80%). **B** As for **A** but for daily COVID-19 deaths. **C** Modelled impact of the BPSV during the first year of the COVID-19 pandemic in different countries around the world, assuming stockpile size varies by World Bank Income Group ("Variable coverage"). Country colour indicates the percentage of COVID-19 deaths occurring in the first year of the pandemic that could have been averted if a BPSV had been available. Bars plot the mean % of deaths averted across

100 simulations, each utilising a single draw from the previously estimated posterior distribution of Rt for each country; error bars represent the 95% confidence interval for those 100 simulations. **D** Modelled impact of the BPSV during the first year of the COVID-19 pandemic in different countries around the world, assuming the variable coverage BPSV scenario−results plotted are the mean of 100 simulations, with country colour indicating the percentage of COVID-19 deaths occurring in the first year of the pandemic that could have been averted if a BPSV had been available. **E** BPSV impact on COVID-19 mortality in Italy during the first year of its COVID-19 epidemic. Grey line indicates model fit to COVID-19 excess mortality data (light grey points), and ribbon indicates the 95% CI across 100 model simulations using different posterior draws for the Rt trajectory. Coloured line indicates expected mortality when the BPSV is available, with the mean trajectory across 100 simulations plotted. Line colours reflect the assumption about the size of the BPSV stockpile. **F** As for **E** but for Iran instead of Italy. **G** As for **F**, but for Bangladesh instead of Italy.

Our results indicate that a BPSV stockpile sufficient to vaccinate 60% of the global eligible population could have averted 54% (37–68%, "Moderate Coverage" scenario) of COVID-19 deaths (Fig. 4A, B), ranging from 40% (21–56 %, "Low Coverage" scenario) to 65% (51–77%, "High Coverage" scenario) depending on the size of the maintained stockpile (Fig. 4C). A global stockpile is unrealistic but could be prioritised in places affected early in the pandemic− assuming nationally maintained stockpiles whose size varied by country averted 50% of deaths (33–65%, "Variable Coverage" scenario, Fig. 4D). Under a "Moderate coverage" scenario, in Italy, BPSV availability could have reduced mortality during the first wave from 1360 daily deaths at its peak to 750 deaths (Fig. 4E) and total COVID-19 mortality over the first year from 124,500 to 59,900 deaths, a 52% reduction (Fig. 4E). Similar impacts were estimated for the epidemics in Iran (Fig. 4F) and Bangladesh (Fig. 4G). Respective stockpile sizes required for these countries to vaccinate their eligible populations and achieve this impact would have been 10.8 million doses for Italy, 5.2 million doses for Iran and 7.9 million doses for Bangladesh.

**BPSV impact depends on target product characteristics**

The presented results make assumptions about the properties of a hypothetical BPSV. The majority of BPSV candidates are at preclinical stages, and their potential properties remain uncertain[17]. We therefore undertook detailed sensitivity analyses to understand how different BPSV properties shape its potential impact, whilst also varying the time taken to develop the more efficacious VSV and features of the pathogen (see Supplementary Table 2 for list of sensitivity analyses). We considered three possible NPI responses to the epidemic: i) Minimal (25% reduction in Rt during BPSV campaign, no NPIs thereafter); ii) Moderate (25% reduction in Rt during BPSV campaign, gradual lifting of NPIs until VSV campaign completes); and iii) Stringent (Rt<1 during BPSV campaign, slow cessation of NPIs until VSV campaign completes). Our results show that increasing BPSV severe disease efficacy reduces mortality in all but the lowest R0 scenarios (Fig. 5A). For low R0 (R0 = 1.5, Fig. S11) and VSV development time of 250 days, BPSV impact is minimal under all NPI scenarios and disease efficacy values considered, with <1.5 deaths per 1000 population averted. Under these scenarios, VSV development is accomplished before significant spread

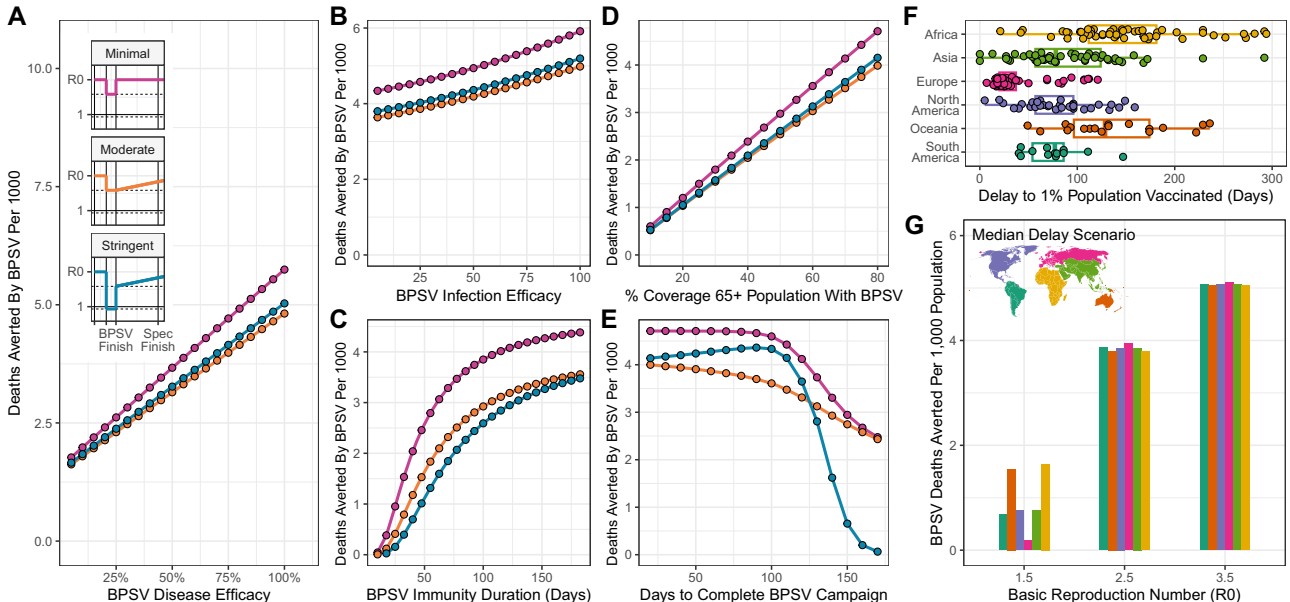

**Fig. 5 | Dependence of BPSV impact on intrinsic vaccine properties, vaccination campaign dynamics and health system capabilities.** Sensitivity analyses exploring the sensitivity of BPSV impact to intrinsic BPSV properties and factors governing the speed, availability and coverage of the BPSV vaccination campaign. **A** Deaths averted by the BPSV (per 1000 population) and BPSV efficacy against severe disease. Results coloured according to NPI scenario considered (pink = minimal, orange = moderate, blue = stringent), for R0 = 2.5. Inset panels show Rt profile for each NPI scenario. Assumed virus-specific vaccine (VSV) development timeline was 250 days. **B** As for (**A**), but for BPSV efficacy against infection. **C** As for (**A**) but for BPSV immunity duration. **D** As for (**A**) but for BPSV stockpile size (and associated coverage of the target population that can be achieved). **E** As for **A**, but

for the rate of vaccination during the BPSV campaign (and the associated time taken to vaccinate all eligible and willing individuals). **F** The delay (in days) between the first country in the world achieving 1% of its population vaccinated with COVID-19 vaccines and other countries achieving this same milestone. Individual coloured points are specific countries—data from Our World In Data. Empirical data are plotted as points, with the underlying boxplot displaying the median (central vertical line), interquartile range (box) and the 1.5× the interquartile range (extent of horizontal whiskers). **G** Impact of delays to BPSV access on deaths averted per 1000 population. Scenarios shown are for moderate NPIs and with continent-specific VSV access delays derived from (**F**).

through the population. For higher R0, increasing BPSV efficacy increases averted mortality (Fig. 5A and Fig. S11A). We observed a less marked influence of infection efficacy on BPSV impact (Fig. 5B and Fig. S11C). This is because the BPSV campaign only targets a small fraction of the population (those aged 60+) and therefore the impact on onwards transmission is limited. BPSV impact increases with longer duration of elicited immunity (Fig. 5C and Fig. S11B).

**BPSV impact is shaped by stockpile size, vaccination campaign speed and access equity**

We additionally carried out analyses exploring how BPSV impact is affected by procurement and health system factors shaping the speed and size of the vaccination campaign. BPSV impact increased linearly with the size of the stockpile (Fig. 5D and Fig. S11D). Assuming an R0 of 2.5, and maintaining a stockpile sufficient to vaccinate 76 % of eligible 60+ (in-keeping with estimates of primary SARS-CoV-2 vaccination coverage of older adults as of December 2022[35]), a BPSV could avert 3.1 deaths per 1000 population. BPSV impact is positively correlated with vaccination campaign speed (Fig. 5E). For the moderate NPI scenario, 3.8 deaths per 1000 are averted when the campaign is completed in <2 months, versus 2.7 deaths per 1000 for a 5-month campaign. However, under the stringent NPI scenario, there was no additional impact of the BPSV if the campaign took more than 5 months. Vaccination campaign speed had even more influence on BPSV impact when R0 = 3.5, and minimal impact when R0 = 1.5 (Fig. S11E).

We also explored the impact of delays to VSV access. During the COVID-19 pandemic, countries in the global south experienced substantial delays before receiving vaccines[9], illustrated by the time taken to achieve 1% vaccination coverage (as a proxy for timeliness of vaccine access)[33,36] (Fig. 5F). Relative to the earliest country vaccinating 1% of their population, European countries experienced a median delay of

32 days (IQR 27-39), whereas for African countries this was 135 days (IQR 110–180). We incorporated these delays into the VSV development timeline and evaluated BPSV impact. For a VSV development time of 250 days, a moderate NPI scenario and R0 = 3.5, the BPSV averted significant mortality, across all vaccine access delays considered (Fig. 5G). However, for R0 = 1.5, the BPSV had a substantially higher impact on disease burden when access was delayed to a level similar to that experienced by the average African (1.75 deaths averted per 1000 population) than the European delay scenario (0.18 deaths averted per 1000 population).

## Discussion

Despite showing promise for disease burden reduction, our results indicate that BPSVs are unlikely to greatly enhance early containment efforts, especially for highly transmissible viruses similar to SARS-CoV-2. A critical issue here is the vaccine delay to protection (VDP, the delay between vaccination and protection), with our results suggesting a negligible fraction of outbreaks contained via ring vaccination when the VDP is of a similar timescale to the generation time of the virus. Given this, other broad-spectrum medical countermeasures such as monoclonal antibodies (where protection arises more rapidly) could be useful additions to the arsenal of tools available for outbreak containment[37,38], as could tools eliciting more robust immunity against infection and onward transmission, such as has been observed for vaccines delivered intranasally[39,40].

Our results did, however, highlight the additional impact that a BPSV can achieve when combined with non-pharmaceutical measures, particularly rapid isolation of cases and quarantine of contacts. In this context, the BPSV can, in several scenarios, act synergistically to push the effective reproduction number below 1 and expand the range of conditions under which control is achievable. Even where suppression

is not attained, deployment consistently slows epidemic growth, a benefit that could result in more limited disease burden (and hence reduced health service burden) ahead of VSV development. Similarly, whilst our results suggest spatially targeted vaccination strategies alone are unlikely to contain a "SARS-CoV-2-like-virus" (except when the area covered by the campaign is large or hospital surveillance is highly sensitive), they highlight the synergistic impacts that can be achieved when combined with quarantine and isolation. An important caveat to these results is the simplifying assumption that omits the strong clustering of infections within households (where secondary-attack rates for SARS-CoV-2 were several-fold higher than in community settings[41]) and the complex network structure over which pathogens spread through populations. Ignoring this structure overestimates the additional benefit that quarantine confers (given that isolation is typically less effective at reducing within-household transmission compared to transmission outside the household), but it is likely to represent a conservative estimate with respect to estimating vaccine impact. This is because household contacts are easier to identify and vaccinate, and additionally, because a branching process without household structure fails to account for the substantial overlap of contacts across successive generations of infections belonging to the same household. Consequently, any fixed stockpile of vaccine doses will cover a larger proportion of the actual at-risk population, increasing the effectiveness of such vaccination strategies.

By contrast, our research shows that utilisation of a BPSV stockpile to reactively vaccinate high-risk groups could reduce both the disease burden and socio-economic impacts of future SARS-X pandemics. During such a pandemic, the availability of a BPSV could, with only limited NPIs, significantly lower mortality to below what could be achieved with longer and more stringent NPIs alone. Such a vaccine therefore has the potential to both reduce disease burden and the wider economic costs associated with NPIs. Indeed, our results suggest that an equitably accessible BPSV stockpile could have averted between 21% and 77% of COVID-19 deaths during the first year of the pandemic. Several important factors modulate this impact. These act at two levels: those intrinsic to properties of the BPSV (which affect the quality of protection it offers, such as efficacy against severe disease) and those that influence the number of individuals infected between BPSV vaccination finishing and the VSV becoming available. It is during this period that the BPSV provides protection to individuals who would otherwise be infected or hospitalised. Faster development timelines for the VSV, or low infection rates (due to stringent NPIs), can diminish the size of the population infected in this period, reducing the impact of the BPSV campaign. Higher transmissibility (R0), limited NPIs and/or longer VSV development times increase the number of individuals infected before the VSV becomes available, increasing the value of a BPSV. Indeed, whilst experiences with the COVID-19 pandemic have motivated significant interest in reducing vaccine development and availability timelines[13], recent work from the Coalition for Epidemic Preparedness Innovations has highlighted that existing approaches to vaccine development are unlikely to yield timelines shorter than 250 days[16]. It is in this context that the BPSV offers significant value, enabling populations to be protected ahead of virus-specific vaccine development and mitigating the limitations associated with reactive vaccine development strategies that are contingent on having identified and sequenced the causative novel pathogen. These results are therefore in keeping with previous work, which has highlighted the likely value of a BPSV-like tool in contexts where the VSV is delayed by >2 months relative to an epidemic's start[25].

An important limitation of this work is that, owing to substantial uncertainty surrounding both the eventual properties of a BPSV and the epidemiological context of a future SARS-X pandemic, our projections are necessarily scenario-based. To capture this uncertainty, we evaluate a broad ensemble of plausible conditions, jointly varying factors relating to the pathogen, the BPSV and the health systems

response. Estimates of BPSV impact therefore span a wide range, reflecting greater or lesser utility depending on the scenario being considered, and highlighting the need for careful consideration of the particular context into which a BPSV is deployed. Indeed, a significant limitation of the results presented relates to uncertainty around BPSV properties. Multiple candidates are under development[17], but no human immunogenicity evaluations have been carried out to date[22,42]. To mitigate this limitation, we assumed BPSV efficacy estimates that are lower than those achieved by mRNA vaccines against ancestral SARS-CoV-2 lineages[43]. We also analysed how the impact of BPSV depends on vaccine properties such as efficacy against disease and vaccine durability. Our results show the BPSV can have a significant impact, especially when VSV development timelines are similar to those achieved for SARS-CoV-2. While the eventual properties of developed BPSVs are uncertain, our results suggest that timely BPSV availability and stockpiling can achieve a significant public health impact, even when efficacy is lower than that of alternative virus-specific vaccines.

An additional limitation throughout the analyses presented here is the assumption that implemented NPIs would have remained unchanged in scenarios with a BPSV. A timely, moderately effective vaccine would likely have influenced both the stringency and duration of NPIs implemented by governments, yet the form and scale of those changes are impossible to reconstruct retrospectively for every country in our dataset. We therefore carried out a suite of sensitivity analyses exploring different NPI scenarios in a hypothetical pandemic scenario. The results reveal a consistent trade-off between deaths averted and NPI days imposed, highlighting that BPSV availability can not only lower mortality but also permit earlier relaxation of population-wide restrictions. Our central estimates should therefore be viewed as conservative with respect to the total public health benefit of a BPSV. If authorities had eased NPIs in response to vaccine deployment, direct mortality reductions might have been smaller, but the net societal benefit would likely have been greater.

A further limitation is that we do not account for existing cross-immunity to a new SARS-X virus. Previous work in SARS-CoV-1 survivors has shown antibodies providing cross-protection to ancestral and variant SARS-CoV-2 virus[21]. Given the global prevalence of SARS-CoV-2, future SARS-X burden (and BPSV impact) may be mitigated by cross-immunity. However, this remains highly uncertain, and indeed, cross-immunity may serve to further increase the impact of a BPSV or expand the range of possible implementation strategies (e.g., as a booster providing more robust immunity against subsequent variant lineages). Additionally, whilst our work highlights the significant public health impact of a BPSV, evaluation of the economic viability of maintaining a stockpile is also required. Currently, this is challenging, due to uncertainty around the eventual properties of developed BPSVs and the cost of acquisition, stockpiling and administration. Whilst previous research has highlighted COVID-19 vaccinations as consistently cost-effective or cost-saving[44], uncertainty in the timing and scale of future outbreaks alongside the costs associated with maintenance of a stockpile necessitates further assessment of the cost-effectiveness of different implementation strategies and mechanisms for stockpiling.

Despite these limitations, our work highlights the significant impact that could be achieved through development, manufacture and stockpiling of BPSVs to facilitate rapid access in a hypothetical SARS-X pandemic. Such vaccines could provide an effective way to protect high-risk groups during the period between novel pathogen identification and the development of efficacious VSVs. In doing so, BPSV utilisation has scope to avert both significant disease burden and substantial economic losses through relaxing the requirement for stringent NPIs to control transmission. However, realising the benefit of these tools will require investment into diagnostics, surveillance and broader public-health response capabilities[45], if they are to effectively form a part of future pandemic preparedness strategies.

## Methods

### Stochastic branching process modelling framework

We extended stochastic branching-process modelling frameworks initially developed to explore SARS-CoV-2 control through contact tracing[46,47] and MERS-CoV vaccination[48] to simulate different vaccination strategies focused on outbreak containment (defined as a final outbreak size of <10,000 infected individuals). These were ring-vaccination (where detection of symptomatic cases triggers reactive vaccination of all contacts of that case, Fig. 1A) and spatially targeted vaccination (where hospitalised cases trigger vaccination of all individuals in a defined geographic area, Fig. 2A). We explicitly model quarantine and isolation through two mechanisms, following approaches to model this developed during the COVID-19 pandemic[31]. These mechanisms are i) contact tracing, in which 47% of contacts identified after their infector becomes symptomatic enter quarantine a mean of 4 days after exposure; and ii) symptom-based self-recognition, in which 90% of individuals isolate a mean of 4 days after their own symptom onset; in both cases quarantine/isolation is assumed to cut onward transmissibility by 65%[31]. See Supplementary Information for full details of the modelling framework.

For both strategies, we calculate the proportion of outbreaks contained relative to a scenario in which the BPSV is not available, whilst varying the pathogen epidemiological properties, the intrinsic properties of the BPSV, and features of the vaccination campaign response. The epidemiological properties that we vary are R0, generation time distribution, extent of pre-symptomatic transmission, proportion of asymptomatic infections and the probability of being hospitalised. We explore two pathogen "archetypes"–the first has properties similar to SARS- 1, with a long generation time, limited pre-symptomatic transmission, low proportion of asymptomatic infections and high disease severity. The second is SARS-CoV-2, with a shorter generation time, extensive pre-symptomatic transmission, more asymptomatic infection and low disease severity. For both archetypes, we vary the R0 across a range of values. Furthermore, we assume that past exposure to SARS-CoV-2 does not generate significant immunity to SARS-X. BPSV properties varied across model runs, including efficacy against infection, relative infectiousness of breakthrough infections, and the assumed delay between vaccination and immunological protection developing.

In all instances, results are the proportion of outbreaks contained across 100 stochastic simulations, per parameter combination and vaccination strategy considered. Code to reproduce the results is available at https://github.com/mrc-ide/diseaseX_modelling. For further details of the model and parameterisation, see Supplementary Information.

### Dynamical compartmental modelling framework

We adapted a published compartmental model of SARS-CoV-2 transmission and vaccination[1] to explore the impact of BPSV availability on disease burden during a future hypothetical SARS-X pandemic. The original model is described in refs. 34,49,50, with details of the extensions added here described below and in the Supplementary Information. Briefly, we extended this modelling framework to enable simulation of two vaccines with distinct properties (the BPSV and the VSV). We model two distinct forms of vaccine efficacy: efficacy against infection and efficacy against severe disease in breakthrough infections. The BPSV, available immediately upon detection, is assumed to have an efficacy of 75% against severe disease and 35% against infection. The disease-specific vaccine, developed later, has a more favourable efficacy profile–95% against severe disease and 55% against infection, with development timelines of either 100 or 250 days. Following spillover, pathogen detection occurs on the first day with ≥5 daily hospitalisations and leads to initiation of both the BPSV vaccination campaign and disease-specific vaccine development. The timing of detection was calculated using the stochastic branching-process

framework. The BPSV is used to vaccinate individuals over the age of 65 years. All age groups except those under 15 are eligible to receive the disease-specific vaccine, with rollout of this vaccine (sufficient to achieve a coverage of 80% of the population) occurring in the oldest age groups first.

### Hypothetical SARS-X pandemic scenario modelling

We assume an R0 of 2.5 and a generation time of 6.7 days, aligned with estimates for the original Wuhan-1 strain, as well as a severity profile and age-specific IFR similar to SARS-CoV-2[51], adjusted to give an overall population-level IFR of 1%. We assume a demographic age-structure matching the age-distribution of the World Bank Upper Middle-Income Country, with the median age, and assume no healthcare constraints that limit the ability of hospitalised individuals to access medical care, noting that relaxing this assumption would increase our estimates of BPSV impact. We assume a rate of BPSV and VSV vaccination equal to the median vaccination rate observed in UMICs during the COVID-19 pandemic (derived from data from Our World In Data[33]). This corresponds to a vaccination rate of 3.5% of the country's population per week and leads to all individuals aged 60+ years being vaccinated within 4 weeks of vaccination beginning. This corresponds to 1.55 million doses per week in a population of 40 million in the representative demography selected (where 15% of the population is aged 60+). We explore different scenarios varying the duration, stringency, and timing of imposed NPIs. These scenarios are each generated from three levels of NPI stringency: i) none, keeping Rt equal to R0; ii) minimal, reducing Rt by 25%; iii) stringent, reducing Rt to 0.9. Using these three levels of NPI stringency, we construct three scenarios for the purposes of analysis: NPI Scenario 1) minimal NPIs applied briefly after pathogen detection until the end of the BPSV campaign; NPI Scenario 2) moderate NPIs lasting until the BPSV campaign's end, then relaxed until the completion of the disease-specific vaccine rollout; NPI Scenario 3) stringent NPIs maintained throughout the BPSV campaign, then eased until the end of the disease-specific vaccine rollout. For each NPI scenario, an "NPI index" (representing a composite of the stringency of imposed NPIs and their duration) was calculated. Deaths averted per 1000 population by the BPSV were estimated by comparing deaths in scenarios with both BPSV and the disease-specific vaccine to scenarios with only the disease-specific vaccine over the first year of the hypothetical pandemic. Potential NPI days averted by the availability of the BPSV were calculated as follows: first, we constructed the Pareto frontier across explored NPI scenarios in the case where only the VSV was available. We then calculated the difference in composite NPI days between the scenario in which the BPSV is available and the composite NPI days for the point lying on the Pareto frontier leading to the same number of deaths as in a scenario where only the VSV is available.

### Retrospective evaluation of potential impact during SARS-CoV-2 pandemic

Using published Bayesian country-specific model fits to excess mortality data[1] we explored the potential impact that a stockpiled BPSV could have had on COVID-19 mortality in the first year of the pandemic. We first sampled 100 draws from the previously estimated posterior distribution of Rt. To then estimate the impact of BPSV, we simulated a counterfactual scenario for each sampled Rt trajectory in which BPSV vaccines were deployed following cumulative globally reported COVID-19 deaths reaching a defined threshold, assuming all countries possess a BPSV stockpile to vaccinate either 40%, 60%, or 80% of their eligible population (low, mid and high BPSV coverage scenarios, respectively) and with rates of BPSV and VSV vaccination specific to each World Bank Income Group and determined based on data from Our World In Data[33]. We also considered an additional coverage scenario determined by World Bank income group strata (20% LIC, 40% LMIC, 60% UMIC, 80% HIC) and a vaccination rate scenario where

global vaccination rate averages were used instead. Deaths averted by the BPSV were calculated by subtracting the estimated COVID-19 deaths from the simulation with the BPSV from the simulation without the BPSV, with the median deaths averted per 1000 population reported here. See Supplementary Information for further details.

## Reporting summary

Further information on research design is available in the Nature Portfolio Reporting Summary linked to this article.

## Data availability

All relevant data and code required to reproduce the analyses presented here are freely available in Github repository (https://github.com/mrc-ide/diseaseX_modelling).

## Code availability

The modelling framework required to carry out the analyses presented here are freely available in Github repositories (https://github.com/mrc-ide/diseaseX_modelling and https://github.com/mrc-ide/squire.page).

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

## Acknowledgements

The investigation was funded by the Coalition for Epidemic Preparedness Innovations (CEPI) through the Vaccine Impact Assessment Modelling project funding. This work was supported by a Sir Henry Wellcome Postdoctoral Fellowship Ref 224190/Z/21/Z. This research was funded in whole, or in part, by the Wellcome Trust (Ref 224190/Z/21/Z). For Open Access, the author has applied a CC BY public copyright licence to any Author Accepted Manuscript version arising from this submission. The work is supported by the MRC Centre for Global Infectious Disease Analysis (reference MR/R015600/1), which is jointly funded by the UK Medical Research Council (MRC), the EDCTP2 programme supported by the European Union and Community Jameel. DJL acknowledges funding from the Wellcome Trust for the Vaccine Impact Modelling Consortium (VIMC) Climate Change Research Programme (grant ID: 226727_Z_22_Z). A.B.H. is supported by an Australian National Health and Medical Research Council Investigator Grant.

## Author contributions

Conceptualization: C.W., G.B., D.O.M., V.C., O.J.W. & A.G. Methodology: C.W., G.B., D.J.L., L.W., O.J.W. & A.G. Investigation: C.W. & A.G. Visualization: C.W., D.O.M., V.C. & A.G. Funding acquisition: O.J.W. & A.G. Project administration: A.G. Supervision: A.G. Writing—original draft: C.W. & A.G. Writing—review & editing: All Authors.

## Competing interests

The Coalition for Epidemic Preparedness Innovations (CEPI) funded the investigation into the impact of the 100 Days Mission. Authors maintained full freedom when designing the study and deciding on additional scenarios to explore. ACG has received personal consultancy fees from HSBC, GlaxoSmithKline, Sanofi and WHO related to COVID-19 epidemiology and from The Global Fund to Fight AIDS, Tuberculosis and Malaria for work unrelated to COVID-19. ACG was previously a non-remunerated member of a scientific advisory board for Moderna and is currently a non-remunerated member of the scientific advisory board for the Coalition for Epidemic Preparedness. OJW has received personal consultancy fees from WHO for work related to malaria. ABH has received personal consultancy fees from WHO for work related to COVID-19, and grant funding for COVID-19 work from WHO and NSW Ministry of Health, Australia. ABH is a member of the WHO Immunization and vaccines related implementation research advisory committee. CW has received personal consultancy fees from SecureBio for work relating to novel pathogen surveillance and from Blueprint Biosecurity for work relating to pandemic preparedness. CWT and LFW and FZ are co-inventors of multiple patent applications on development of pan-sarbecovirus vaccines and human-nAbs. All other authors declare no competing interests.

## Additional information

Charles Whittaker [1,2,3] ✉, Gregory Barnsley [4], Daniela Olivera Mesa [1], Victoria Cox [1], Daniel J. Laydon [1,5], Chee Wah Tan [6,7], Feng Zhu [7], Rob Johnson[1,8], Patrick Doohan [1,8], Lilith K. Whittles [1], Gemma Nedjati-Gilani[1], Peter Winskill [1], Alexandra B. Hogan[1,9], Arminder Deol [10], Christinah Mukandavire[10], Katharina Hauck [1,8], David Chien Boon Lye[11,12,13,14], Lin-Fa Wang [6,15], Oliver J. Watson [1,8] & Azra C. Ghani [1,13,16] ✉

[1]MRC Centre for Global Infectious Disease Analysis, School of Public Health, Imperial College London, London, UK. [2]Division of Infectious Diseases and Vaccinology, School of Public Health, University of California, Berkeley, CA, USA. [3]Center for Computational Biology, College of Computing, Data Science, and Society, University of California, Berkeley, CA, USA. [4]London School of Hygiene and Tropical Medicine, London, UK. [5]Centre for Health Economics & Policy Innovation, Department of Economics & Public Policy, Imperial College Business School, London, UK. [6]Programme in Emerging Infectious Diseases, Duke-NUS Medical School, Singapore, Singapore. [7]Infectious Diseases Translational Research Programme, Yong Loo Lin School of Medicine, National University of Singapore, Singapore, Singapore. [8]Abdul Latif Jameel Institute for Disease and Emergency Analytics, School of Public Health, Imperial College London, London, UK. [9]School of Population Health, Faculty of Medicine and Health, University of New South Wales, Sydney, NSW, Australia. [10]Coalition for Epidemics Preparedness Innovations, Oslo, Norway. [11]National Centre for Infectious Diseases, Singapore, Singapore. [12]Department of Infectious Diseases, Tan Tock Seng Hospital, Singapore, Singapore. [13]Lee Kong Chian School of Medicine, Nanyang Technological University, Singapore, Singapore. [14]Yong Loo Lin School of Medicine, National University of Singapore, Singapore, Singapore. [15]SingHealth Duke-NUS Global Health Institute, Singapore, Singapore. [16]Saw Swee Hock School of Public Health, National University of Singapore, Singapore, Singapore. ✉e-mail: cwhittaker@berkeley.edu; a.ghani@imperial.ac.uk

