## [Transparent Peer Review file · Nature Communications]

Quantifying the impact of a broadly protective sarbecovirus vaccine in a future SARS-X pandemic

Corresponding Author: Dr Charlie Whittaker

Version 0:

Reviewer comments:

Reviewer #1

(Remarks to the Author)

The authors use mathematical models to explore the potential impact of a broadly protective sarbecovirus vaccine (BPSV) in a future SARS-X pandemic. They employ branching processes to simulate ring and spatial vaccination strategies for two hypothetical viruses SARS-CoV-1-like and SARS-CoV-2-like. Sensitivity analyses of vaccine properties, epidemic dynamics, and healthcare capacity show that while BPSV strategies may not fully contain a SARS-CoV-2-like virus, they could slow or contain a SARS-CoV-1-like virus. They then use a compartmental modelling framework to evaluate the impact of BPSV prioritizing the elderly population under varying levels of NPIs in a SARS-X outbreak. They quantify the effects of BPSV by comparing the number of deaths in the simulations with/without BPSV. Furthermore, they analyse how stockpiled BPSV could have reduced mortality during the first year of the COVID-19 pandemic. Their findings indicate that stockpiled BPSV could have lowered deaths significantly.

The work is substantial, and the topic is of clear interest, however, there are several issues that need to be considered and addressed before publication.

Major concerns

The authors present a monumental amount of work, considering many different scenarios, modelling approaches, and assumptions in a few pages. They clearly opted for a high-level descriptive narrative, which probably is necessary being mindful of length, however, do we really need a full compendium on the matter in a single paper? Sometimes less is more. It might be worth to have a discussion with the editor and focus the paper on 1-2 points, but providing more details and a more careful discussion about the assumptions made and how these affect the results.

Connecting to this point, the modelling framework is complicated with many assumptions and parameters. Though the authors provide descriptive explanations, the models' details are far from clear. The authors should provide more details (including their mathematical structure) about each model used instead of pointing the reader to other papers.

The authors use different methods for modelling ring/spatial-targeted strategies and high-risk population strategies of BPSV. They also evaluate the impact of vaccine strategies from different aspects (containing a pandemic for ring/spatial, reducing mortality for vaccination of high-risk population). However, the context behind these strategies is unclear. Could the authors clarify why they propose different strategies and why they approach them differently? Further explanation of the rationale and assumptions for each strategy would enhance understanding.

The authors do not mention whether previous studies have modelled or assessed the impact of BPSV. They should investigate existing literature to determine if their work is the first on this topic. If similar research exists, they should provide a comparison to show how their work builds on or differs from previous studies.

The authors use branching processes to model ring/spatial vaccination strategies and a compartmental model for mass population vaccination. These are two different techniques. However, it is unclear whether these two approaches are consistent. For example, in sub-section 1.1 of the supplementary file, they assume generation and incubation times follow a Gamma distribution in branching processes, but they do not mention what the case is in the compartmental model.

Additionally, as mentioned in question 1, all models should be presented mathematically in detail to improve transparency.

In their evaluation of the potential impact of BPSV on the COVID-19 Pandemic, the authors consider, effectively, a counterfactual scenario comparing the outcomes of a model calibrated to real deaths with the same model featuring BPSV. This is standard practice; however, they do not mention how this implies assuming same levels of NPIs in the two cases. Is this realistic? How does this change the projections? On this point the authors mentioned that they get the calibrated projections from published work, however the reference cited is a pre-preprint and the reader is left with the only choice to read another paper to understand with some level of confidence how these fits are generated. As mentioned above, this is far from optimal and should be avoided.

On this point, would have BPSV changed (re)importation dynamics across and within countries? This might have played a key role especially in the first phases. It would be great to explore this point, maybe in future work, and for now potentially mention how the counterfactual scenario does not account for this.

On the same main theme, more information is required to describe the how spatially targeted vaccinations are implemented. Again, the reader is left on the dark about the details and provided just a high-level description. This is not a publication for dissemination, hence all details, especially for reproducibility, are key.

In all the different models and scenario considered the authors try different assumptions, run large sensitivity analyses, yet they pick a few (arguably arbitrary) main cases as result. However, in practice the real uncertainty of the actual impact of BPSV is high. Hence, is it possible to present all the results conveying this uncertainty more explicitly?

In several cases the author implement rollouts of BPSV (as well as virus specific vaccines) providing absolute numbers which however are hard to grasp nor easy to evaluate. For example, in line 150, the authors speak about 1.55 million doses per week. It would be important to put all these numbers and rates into context. How do they compare with what we saw in the early rollout of covid-19 vaccines, or other vaccination campaigns?

Going back to uncertainty, it would be important to add confidence intervals in the various line plots. For example, panels H and I in figure 1, panels C and D in figure 2, panels E, F, G in figure 3.

At the end of the first paragraph of the discussions the authors say "...with the exact magnitude of deaths averted depending on the size of the BPSV stockpile." While this is true, there are so many other factors (as manifested by the large sensitivity analysis conducted) that affect the results. Again, the reader should have in mind the real uncertainty of the results presented, including the many (really many) assumptions made.

Surprisingly, despite the many assumptions, unknowns, and possible non-linear interactions between them, the authors do not report any limitations of the different models used in the discussions.

The SARS-X Pandemic sounds very similar to SARS-CoV-2, in terms of the choice of several features of the virus. A motivation of those choices would be helpful.

Minor points

Is there any reference for spatial-targeted vaccination strategy? The authors show the reference for ring strategy but do not mention if the spatial one is first proposed in their work or from previous work.

In the simulation of vaccinating high-risk populations during a future SARS-X outbreak, the authors calculate averted deaths by BPSV, but they do not specify the time window for this calculation. Please clarify the period over this.

In Figures 3E-3G, the fraction of averted deaths at the peak in Bangladesh is significantly higher compared to Italy and Iran. Could the authors investigate and explain the factors contributing to this difference in averted deaths across countries? Additionally, in Figure 4E, under stringent conditions, the impact of vaccination increases slightly as the vaccine rollout rate decreases, which is unexpected. Could the authors explain further about it?

In the hypothetical SARS-X pandemic scenario, the rate at which BPSV is administered is unclear. The authors should provide the details about this.

In the retrospective evaluation of BPSV's impact on the COVID-19 pandemic, the authors assume that BPSV is triggered by a threshold of cumulative deaths. What is the rationale behind this threshold, and how was it determined? Additionally, in the hypothetical SARS-X scenario, BPSV is triggered by pathogen detection (as shown in the supplementary material). Why do the authors use different triggers (cumulative deaths vs. pathogen detection) in these two scenarios? Please explain the reasoning for these different settings.

When discussing the results of containment of SARS-1-like and SARS-2-like viruses, the role of asymptomatic transmission should be made more apparent.

The authors write "to simulate different vaccination strategies focussed on outbreak containment (defined as a final outbreak size of <10,000 infected individuals)." Where this threshold comes from?

There are some typos in the manuscript:

1) On the line 197-198, the figure labels seem wrong. For Italy, Iran, and Bangladesh, the labels should be Fig 3E, 3F, 3G respectively instead of 3D, 3E, 3F. Please check the labels in the whole paragraph.

2) On the line 198, there should be a space between 'from' and '124,500'.

3) On the line 207, '...also varying the time the time...', 'the time' repeats twice.

(Remarks on code availability)

The codes are provided in R. As I don't really know R I cannot judge the code provided. Nevertheless, the github page is well organized.

Reviewer #2

(Remarks to the Author)

(Remarks on code availability)

Reviewer #3

(Remarks to the Author)

The paper presents a modelling study to investigate the potential impact of stockpiled broadly protective vaccine (with reduced efficacy comparatively to a targeted vaccine) in reducing disease burden in the event of a future SARS-X outbreak. The paper is well presented, and with its focus on pandemic preparedness, timely.

The first sections, concerned with using a branching process model to assess the potential for viral containment, were the weakest both in terms of approach and relevance of the results. As a containment strategy, ring vaccination seems to lack plausibility. During the covid-19 pandemic, where early cases were detected and contacts identified, containment was attempted universally through the use of isolation (which is theoretically 100% efficacious against infection). Given that this was still only partially successful, the idea of replacing isolation of contacts in the early stages of a pandemic with a low efficacy vaccine seems unrealistic.

Combining existing isolation strategies with spatially targeted BPSV in the early stages of an outbreak may have significant value however. The results presented demonstrate that both spatially targeted and ring vaccination are unlikely to contain an epidemic, even when it results from an identified index case, however it would be valuable to know how effective a spatially targeted BPSV campaign would be in slowing an early outbreak (effect on R_0 ?). BPSV use in high risk areas for importation (or in the locality of identified early cases) may have significant benefits in slowing down early outbreaks, increasing the chances of containment through case/contact isolation, and providing more time to develop other counter measures.

My suggestion would be to reimplement the branching process model to reflect this application; lose the focus on ring vaccination and instead show how effective spatially targeted BPSV may be in reducing R_0 in the early stages of an outbreak, and discuss the benefits this may bring.

If left in their current presentation, results from the branching process model also seem to be misleading in presenting a comparison between two different vaccination strategies. Each strategy relies on a completely different set of assumptions and tested for different sensitivities. For instance the spatial strategy is triggered upon a hospitalisation threshold (meaning a potentially large number of initial cases) and ring vaccination is more ambitiously triggered upon a symptomatic index case.

Finally, where the branching process model is used, it would be good to see some discussion of the limitations of the model. It may be an effective simplification in the early stages of an outbreak, but given the weight of evidence to support the importance of household infections within the covid pandemic, the branching model is likely insufficient to capture the true network structure and it is important to understand how this may affect the results presented.

The remainder of the results were based on a compartmental covid model fitted to 185 countries. This model has already had substantial use elsewhere. The results from this model were generally compelling and well presented.

My main criticism here would be the assumption that all countries manage an equally fast BPSV deployment. While I strongly support the importance of showing equity in global vaccine coverage and availability, we are a long way off being able to have the infrastructure to deploy vaccines equally fast across the globe; this is due to challenges not just in resources, but also infrastructure, population structure, and environmental factors. The reality then could significantly impact results, and this would ideally be taken into account in the model.

Finally, while I understand the desire for brevity and avoiding repeating details shown elsewhere. I would've found more model details helpful throughout, especially in the supplement. In particular detailing all the parameters in the model, where they come from, how they're fitted, which ones differ between countries, where does uncertainty come from in the model?

A Fig 5 is referred to in the supplement, but does not exist, can you check this label?

In Fig 2c (and similar) can you put the crosses in front on the dots, so it is obvious whether they are actually the same or just absent on some points?

(Remarks on code availability)

Version 1:

Reviewer comments:

Reviewer #1

(Remarks to the Author)

The author did a good job revising the manuscript and addressing most points raised in the first round of review. A few points however need still attention:

In the revised version, explicit mathematical equations describing the epidemic dynamics are not included. The authors opted to cite a reference for the model. However, upon review, that reference also lacks mathematical equations and instead points to another source. Despite being published somewhere, it would be important to add these details of the model in this paper (of course in the SI) to make the publication self-contained.

Regarding the uncertainty, the authors state that data fitting was only performed for the retrospective SARS-CoV-2 in Figure 4. Nevertheless, confidence intervals should also be provided for the low, moderate, and high coverage scenarios in Figure 4 panels (E), (F), and (G), in addition to the original no-BPSV case.

In figure 4, the description for panel (C) is missing, and the labels currently designated as (C), (D), and (E) should be shifted to correspond to panels (D), (E), and (F), respectively.

(Remarks on code availability)

I don't really know R hence I cannot comment on the details of the code. The github looks well organized though

Reviewer #2

(Remarks to the Author)

(Remarks on code availability)

The authors have provided a clear README file with instructions for installation and execution in the repository.

Reviewer #3

(Remarks to the Author)

My thanks to the authors for taking time to consider my original feedback, they have done a great job of addressing my concerns and suggestions. The revised manuscript appears much more robust, relevant, and applicable. I only have a few very minor new comments.

In figure's 1 and 2, can sensitivity to lower vaccine efficacy also be shown? I.e. Given an assumed vaccine efficacy of 35% it would seem more sensible to present the heatmaps in fig.1/2G to span between 0-70% unless there is good reason that higher efficacy vaccines are likely?

Also, in these heatmaps, I assume the reference scenario looks different between scenarios due to stochasticity? (could the same simulation set be used for each of these?). How many simulations are used for these. In all the results it would be beneficial to present the effects of stochasticity more (prediction intervals shown on lines/bars when it doesn't negatively impact readability). The confidence intervals in 4E-F seem very narrow, though this may well be due to model limitations (dynamics that are not considered).

It is now stated in the limitations that using a branching process model, rather than the true network structure may underestimate vaccine reach since "household contacts are easier to identify and vaccinate". Perhaps more significantly, it should be said that a branching model underestimates vaccination impact since the it overestimates the size of the exposed population (generation n will feature many of the same individuals already seen in generations 1 to n-1).

(Remarks on code availability)

Na

REVIEWER COMMENTS

Reviewer #1 (Remarks to the Author):

The authors use mathematical models to explore the potential impact of a broadly protective sarbecovirus vaccine (BPSV) in a future SARS-X pandemic. They employ branching processes to simulate ring and spatial vaccination strategies for two hypothetical viruses SARS-CoV-1-like and SARS-CoV-2-like. Sensitivity analyses of vaccine properties, epidemic dynamics, and healthcare capacity show that while BPSV strategies may not fully contain a SARS-CoV-2-like virus, they could slow or contain a SARS-CoV-1-like virus. They then use a compartmental modelling framework to evaluate the impact of BPSV prioritizing the elderly population under varying levels of NPIs in a SARS-X outbreak. They quantify the effects of BPSV by comparing the number of deaths in the simulations with/without BPSV. Furthermore, they analyse how stockpiled BPSV could have reduced mortality during the first year of the COVID-19 pandemic. Their findings indicate that stockpiled BPSV could have lowered deaths significantly.

The work is substantial, and the topic is of clear interest, however, there are several issues that need to be considered and addressed before publication.

Major concerns

The authors present a monumental amount of work, considering many different scenarios, modelling approaches, and assumptions in a few pages. They clearly opted for a high-level descriptive narrative, which probably is necessary being mindful of length, however, do we really need a full compendium on the matter in a single paper? Sometimes less is more. It might be worth to have a discussion with the editor and focus the paper on 1-2 points, but providing more details and a more careful discussion about the assumptions made and how these affect the results.

We appreciate the reviewer's feedback regarding the breadth of the manuscript. Our aim with this work was to provide a comprehensive analysis of BPSV impact across various scenarios, reflecting the diversity of possible use cases for a BPSV during a pandemic. This diversity of possible use-cases (and their comparative merits/limitations) we believe supports the inclusion of them all in a single manuscript. We recognise however that the use of multiple different approaches has the potential to be confusing. We have therefore modified the contents of the paper to better clarify and delineate the different use-cases (including splitting out the ring/spatially-targeted vaccination approaches into their own figures) and modelling approaches utilised and updated both figures and text to better convey to readers the uncertainty and assumptions underpinning the analyses presented in this work.

Connecting to this point, the modelling framework is complicated with many assumptions and parameters. Though the authors provide descriptive explanations, the models' details are far from clear. The authors should provide more details (including their mathematical structure) about each model used instead of pointing the reader to other papers.

Thanks for this feedback. We agree and based on this suggestion have added additional further information on the modelling frameworks in the Supplementary Information, so

that readers are able to better discern and understand what we have done. We note that the compartmental model utilised has been previously published and detailed extensively here: <https://www.sciencedirect.com/science/article/pii/S0264410X21004278>. We therefore focus the technical details in the manuscript on the way in which this model is utilised, and the extension on the representation of vaccination required to simulate the BPSV. For the branching-process framework, which hasn't previously been published, we have added significant further information describing the technical and algorithmic details of the implementation for both ring-vaccination and spatially targeted vaccination strategies.

The authors use different methods for modelling ring/spatial-targeted strategies and high-risk population strategies of BPSV. They also evaluate the impact of vaccine strategies from different aspects (containing a pandemic for ring/spatial, reducing mortality for vaccination of high-risk population). However, the context behind these strategies is unclear. Could the authors clarify why they propose different strategies and why they approach them differently? Further explanation of the rationale and assumptions for each strategy would enhance understanding.

Thanks for this feedback. We agree and have added substantial further information on the modelling frameworks (and their mathematical structure) utilised in this work to the Supplementary Information, so that readers are able to better discern and understand what we have done. Modelling of these different strategies was directly informed through conversations with scientists developing BPSVs and organisations actively supporting the development of BPSVs. The reason for approaching these strategies differently from a methodological perspective is rooted in their distinct focus on different epidemic stages.

Containment strategies, such as ring and spatial-targeted vaccination, aim to interrupt transmission during the early, stochastic phase of an outbreak. During this phase, the small number of cases and potential for high variability in outcomes make deterministic compartmental models inappropriate. Instead, we use stochastic models. Branching processes are appropriate to capture these early dynamics as they represent the contact patterns and stochastic nature of early stages of an outbreak. However, as they do not capture herd immunity effects, we subsequently switch to using compartmental models to explore the impact on epidemics that are not contained. Strategies targeting disease burden reduction (e.g., vaccinating high-risk populations) are more relevant in the later stages of an epidemic, where the outbreak has grown larger and population immunity is building. At this stage, deterministic compartmental models are computationally efficient and appropriate for evaluating the population-level impact of interventions. We have added text in the Supplementary Information clarifying this distinction to help readers better understand the rationale for employing different modelling frameworks for these strategies.

The authors do not mention whether previous studies have modelled or assessed the impact of BPSV. They should investigate existing literature to determine if their work is the first on this topic. If similar research exists, they should provide a comparison to show how their work builds on or differs from previous studies.

Thanks for raising this point. Our searches of the literature have revealed a dearth of modelling focussed on understanding the potential epidemiological impact of broad-

spectrum medical countermeasures at a population-level. The limited prior work that exists has been focussed on other virus families (e.g. influenza: <https://www.sciencedirect.com/science/article/pii/S1755436524000379>) or limited in its use case and geographical focus (e.g. USA and disease burden reduction: [https://www.thelancet.com/journals/eclinm/article/PIIS2589-5370\(23\)00546-1/fulltext](https://www.thelancet.com/journals/eclinm/article/PIIS2589-5370(23)00546-1/fulltext)). We agree with the reviewer this work is important to mention however, and have added text reflecting this prior literature to both the Introduction and Discussion.

Introduction: “Previous modelling studies focused on the US have highlighted that a stockpiled pan-coronavirus vaccine could avert significant disease burden and be cost-saving, even in the context of modest vaccine efficacy and partial rollout²⁵. Complementary work evaluating hypothetical universal influenza A vaccines reaches similar conclusions, underscoring the importance of breadth, durability and speed of deployment²⁶.”

Discussion: “It is in this context that the BPSV offers significant value, enabling populations to be protected ahead of virus-specific vaccine development and mitigating the limitations associated with reactive vaccine development strategies that are contingent on having identified and sequenced the causative novel pathogen. These results are therefore in-keeping with previous work which has highlighted the likely value of a BPSV-like tool in contexts where the VSV is delayed by >2 months relative to an epidemic’s start²⁵.”

The authors use branching processes to model ring/spatial vaccination strategies and a compartmental model for mass population vaccination. These are two different techniques. However, it is unclear whether these two approaches are consistent. For example, in sub-section 1.1 of the supplementary file, they assume generation and incubation times follow a Gamma distribution in branching processes, but they do not mention what the case is in the compartmental model. Additionally, as mentioned in question 1, all models should be presented mathematically in detail to improve transparency.

We have attempted to keep the two approaches consistent in terms of parameterisation as far as is possible within the different modelling frameworks. To do so, the compartmental modelling framework utilises two compartments for each delay distribution which results in an Erlang-2 distribution (a subset of Gamma distribution).

In their evaluation of the potential impact of BPSV on the COVID-19 Pandemic, the authors consider, effectively, a counterfactual scenario comparing the outcomes of a model calibrated to real deaths with the same model featuring BPSV. This is standard practice; however, they do not mention how this implies assuming same levels of NPIs in the two cases. Is this realistic? How does this change the projections? On this point the authors mentioned that they get the calibrated projections from published work, however the reference cited is a pre-preprint and the reader is left with the only choice to read another paper to understand with some level of confidence how these fits are generated. As mentioned above, this is far from optimal and should be avoided.

*The reviewer raises an important point. It is impossible for us to predict how availability of the BPSV would have changed patterns of implemented NPIs, especially across the diverse range of countries we model (and the reviewer well notes that our assumption of unchanged NPIs is standard practice). We do however recognise this is a significant assumption, and so addressed this in part in **Figure 3**, where we explored BPSV impact across a range of NPI scenarios in a hypothetic pandemic. Moreover, these analyses*

highlight a trade-off between deaths averted and NPI days averted (Figure 3C and 3D) - were NPIs to be changed in response to the availability of a BPSV, this would be another benefit of the BPSV (enabling fewer NPIs to be implemented and thus minimising their associated socio-economic damage). Given the uncertainty in how these scenarios would play out in a global pandemic however, we chose to not explore this – but have added text to the Discussion highlighting this nuanced point in more detail:

Discussion: *“An additional limitation throughout the analyses presented here is the assumption that implemented NPIs would have remained unchanged in scenarios with a BPSV. A timely, moderately effective vaccine would likely have influenced both the stringency and duration of NPIs implemented by governments, yet the form and scale of those changes are impossible to reconstruct retrospectively for every country in our dataset. We therefore carried out a suite of sensitivity analyses exploring different NPI scenarios in a hypothetical pandemic scenario. The results reveal a consistent trade-off between deaths averted and NPI days imposed, highlighting that BPSV availability can not only lower mortality but also permits earlier relaxation of population-wide restrictions. Our central estimates should therefore be viewed as conservative with respect to the total public health benefit of a BPSV. If authorities had eased NPIs in response to vaccine deployment, direct mortality reductions might have been smaller, but the net societal benefit would likely have been greater.”*

Regarding the reviewer’s 2nd point, we agree with their concerns about citing pre-prints and apologise for the oversight. The cited pre-print has now been published (see here: [https://www.thelancet.com/journals/langlo/article/PIIS2214-109X\(24\)00286-9/fulltext](https://www.thelancet.com/journals/langlo/article/PIIS2214-109X(24)00286-9/fulltext)) and we have updated the citation to reflect that this work and the associated fits have passed through a rigorous, peer-reviewed process of evaluation.

On this point, would have BPSV changed (re)importation dynamics across and within countries? This might have played a key role especially in the first phases. It would be great to explore this point, maybe in future work, and for now potentially mention how the counterfactual scenario does not account for this.

This is a good point – it’s possible the BPSV might have changed (re)importation dynamics. In practice, we don’t expect this effect to be large i) because the BPSV is distributed to age-groups who contribute only a relative small amount to transmission, especially internationally (i.e. elderly individuals) and ii) we model only a small, limited BPSV effect on infection and onwards transmissibility. We indirectly explore the impact of changes to (re)importation dynamics by varying the assumed global cumulative mortality toll at which the BPSV stockpile is activated – with our results showing minimal impact of changes to this timing on BPSV impact.

On the same main theme, more information is required to describe the how spatially targeted vaccinations are implemented. Again, the reader is left on the dark about the details and provided just a high-level description. This is not a publication for dissemination, hence all details, especially for reproducibility, are key.

Thanks for flagging that this is unclear – we have extended and added to the description of the spatially targeted vaccination implementation in the supplementary information, so that the reader might have more clarity on the approach. We also note that from a reproducibility perspective, code for implementing the spatially targeted vaccination

approach is available here: https://github.com/mrc-ide/diseaseX_modelling/blob/main/functions/spatial_vax_simulation.R

In all the different models and scenario considered the authors try different assumptions, run large sensitivity analyses, yet they pick a few (arguably arbitrary) main cases as result. However, in practice the real uncertainty of the actual impact of BPSV is high. Hence, is it possible to present all the results conveying this uncertainty more explicitly?

*We note that because we are not fitting to data in any analyses (except for the SARS-CoV-2 retrospective impact figure), the presentation of uncertainty intervals (which are predicated on distributional assumptions about the comparative likelihood of different parameter values) are inappropriate here and would actually mislead readers, giving them a sense of certainty about expectation and range of values in a way that is not statistically principled. Instead, we convey uncertainty, as the reviewer notes, through carrying out a comprehensive suite of sensitivity analyses – sensitivity analyses which we have extended for the purpose of this manuscript review (see **new figures S1-S8**). We have also added text to the Discussion better highlighting the fact that the estimates presented here are dependent on assumptions of certain epidemic/pandemic scenarios, and the impact of these assumptions on the presented results:*

Discussion: *“An important limitation of this work is that, owing to substantial uncertainty surrounding both the eventual properties of a BPSV and the epidemiological context of a future SARS-X pandemic, our projections are necessarily scenario-based. To capture this uncertainty, we evaluate a broad ensemble of plausible conditions, jointly varying factors relating to the pathogen, the BPSV and the health systems response. Estimates of BPSV impact therefore span a wide range, reflecting greater or lesser utility depending on the scenario being considered, and highlighting the need for careful consideration of the particular context into which a BPSV is deployed.”*

In several cases the author implement rollouts of BPSV (as well as virus specific vaccines) providing absolute numbers which however are hard to grasp nor easy to evaluate. For example, in line 150, the authors speak about 1.55 million doses per week. It would be important to put all these numbers and rates into context. How do they compare with what we saw in the early rollout of covid-19 vaccines, or other vaccination campaigns?

We note that the vaccination campaigns simulated in the analyses presented in Figure 3 (now Figure 4) are based on rates derived from Our World In Data’s COVID-19 vaccination data (https://ourworldindata.org/grapher/cumulative-covid-vaccinations?country=OWID_AFR~OWID_EUR) i.e. they are empirically motivated and largely similar to the rates achieved during the COVID-19 vaccination campaigns. We understand the reviewer’s point however and have therefore added text to both the Results and Methods sections that further contextualise these results.

Methods: *“We assume a rate of BPSV and VSV vaccination equal to the median vaccination rate observed in UMICs during the COVID-19 pandemic (derived from data from Our World In Data³³). This corresponds to a vaccination rate of 3.5% of the country’s population per week and leads to all individuals aged 60+ year being vaccinated within 4 weeks of vaccination beginning. This corresponds to 1.55 million doses per week in a population of 40 million in the representative demography selected (where 15% of the population are aged 60+).”*

Results: “We illustrate impact with vaccination rates of 3.5% of the country’s population per week, such that the 60+ age-group is vaccinated within 4 weeks (Fig 3A) – this corresponds to 1.55 million doses per week in a population of 40 million in the representative demography selected (where 15% of the population are aged 60+), and is derived from vaccination rates observed in UMICs during the COVID-19 pandemic³³”

Going back to uncertainty, it would be important to add confidence intervals in the various line plots. For example, panels H and I in figure 1, panels C and D in figure 2, panels E, F, G in figure 3.

As above, we note that because we are not fitting to any data in these analyses, the presentation of confidence intervals (which are predicated on distributional assumptions about the comparative likelihood of different parameter values) are inappropriate here. Instead, we convey uncertainty, as the reviewer notes above, through carrying out a comprehensive suite of sensitivity analyses.

At the end of the first paragraph of the discussions the authors say “...with the exact magnitude of deaths averted depending on the size of the BPSV stockpile.” While this is true, there are so many other factors (as manifested by the large sensitivity analysis conducted) that affect the results. Again, the reader should have in mind the real uncertainty of the results presented, including the many (really many) assumptions made.

We agree with the reviewer, and have removed this particular piece of text, per the reviewer’s suggestion. We have now combined that initial paragraph with the following one which discusses in much more detail the different factors shaping BPSV impact, and note also that our updated Discussion text includes much around how not just uncertainties in BPSV intrinsic properties (such as efficacy, which was discussed previously) but other additional factors (such as assumptions around implemented NPIs) will determine BPSV impact.

Surprisingly, despite the many assumptions, unknowns, and possible non-linear interactions between them, the authors do not report any limitations of the different models used in the discussions.

See above – we have added text in both the Discussion and the Supplementary Information highlighting the limitations of our approach and the assumptions that underpin the results presented here.

The SARS-X Pandemic sounds very similar to SARS-CoV-2, in terms of the choice of several features of the virus. A motivation of those choices would be helpful.

Our motivation was that the recency and comparatively data-rich setting of the COVID-19 pandemic provides a substantive and empirically motivated example in which to explore impact. Moreover, SARS-CoV-2 is a member of the sarbecoviruses and thus represents a useful starting point for analyses exploring the potential impact of a BPSV. We do of course note however that hypothetical future SARS-X pandemics might be driven by pathogens with properties different to SARS-CoV-2. That was behind our motivation for exploring a pathogen with properties similar to SARS-CoV in our initial analyses of ring-vaccination and spatially targeted vaccination strategies, and behind our motivation for exploring a range of basic reproduction numbers (as a proxy for transmissibility) in the subsequent analyses of the disease burden reduction use case.

Minor points

Is there any reference for spatial-targeted vaccination strategy? The authors show the reference for ring strategy but do not mention if the spatial one is first proposed in their work or from previous work.

The WHO Strategic Advisory Group of Experts (SAGE) released guidance offering updates and adaptations to traditional ring-vaccination approaches that included targeted geographical vaccination (see here: <https://www.who.int/news/item/23-09-2019-second-ebola-vaccine-to-complement-ring-vaccination-given-green-light-in-drc>). That was our motivation for including the strategy, though note we did not cite this document in the initial version of the manuscript – we have now added that reference in.

***Results:** “We explored the impact of spatially-targeted vaccination strategies utilising the BPSV for containment, similar to the approaches that have recently been utilised during ebola outbreaks²⁹.”*

In the simulation of vaccinating high-risk populations during a future SARS-X outbreak, the authors calculate averted deaths by BPSV, but they do not specify the time window for this calculation. Please clarify the period over this.

***Methods:** “Deaths averted per 1,000 population by the BPSV were estimated by comparing deaths in scenarios with both BPSV and the disease-specific vaccine to scenarios with only the disease-specific vaccine over the first year of the hypothetical pandemic.”*

In Figures 3E-3G, the fraction of averted deaths at the peak in Bangladesh is significantly higher compared to Italy and Iran. Could the authors investigate and explain the factors contributing to this difference in averted deaths across countries? Additionally, in Figure 4E, under stringent conditions, the impact of vaccination increases slightly as the vaccine rollout rate decreases, which is unexpected. Could the authors explain further about it?

Bangladesh’s epidemic occurs later than the epidemics in Italy and Iran (these were some of the earliest epidemics during the pandemic), and therefore a greater fraction of Bangladesh’s eligible population is protected with the BPSV before being infected.

The author raises a good point about 4E and stringent NPI conditions. This relates to the timing of NPI release under the stringent scenario. In this scenario, “stringent NPIs” (i.e. NPIs that reduce R below 1) are implemented until the end of the BPSV campaign – when the BPSV takes longer to be delivered, a longer and more stringent NPI campaign is in place. The suppression of community infection prevalence that occurs because of this slightly outweighs the impact of faster BPSV delivery (up to ~100 days to complete BPSV campaign), which is why a slight increase is observed. The complexities of the interactions between NPIs and campaign speed are part of the reason we present a suite of sensitivity analyses exploring impact across a range of possible NPI scenarios.

In the hypothetical SARS-X pandemic scenario, the rate at which BPSV is administered is unclear. The authors should provide the details about this.

We have added text to both the Methods, Results and Supplementary Information clarifying the rate at which the BPSV is administered, and the source this was derived

from (Our World In Data). Alongside suggestions from Reviewer 3, we have also added a sensitivity analysis (**Figure S10**) relating to this point.

Methods: “We assume a rate of BPSV and VSV vaccination equal to the median vaccination rate observed in UMICs during the COVID-19 pandemic (derived from data from Our World In Data³³). This corresponds to a vaccination rate of 3.5% of the country’s population per week and leads to all individuals aged 60+ year being vaccinated within 4 weeks of vaccination beginning. This corresponds to 1.55 million doses per week in a population of 40 million in the representative demography selected (where 15% of the population are aged 60+).”

Results: “We illustrate impact with vaccination rates of 3.5% of the country’s population per week, such that the 60+ age-group is vaccinated within 4 weeks (**Fig 3A**) – this corresponds to 1.55 million doses per week in a population of 40 million in the representative demography selected (where 15% of the population are aged 60+), and is derived from vaccination rates observed in UMICs during the COVID-19 pandemic³³”

In the retrospective evaluation of BPSV's impact on the COVID-19 pandemic, the authors assume that BPSV is triggered by a threshold of cumulative deaths. What is the rationale behind this threshold, and how was it determined? Additionally, in the hypothetical SARS-X scenario, BPSV is triggered by pathogen detection (as shown in the supplementary material). Why do the authors use different triggers (cumulative deaths vs. pathogen detection) in these two scenarios? Please explain the reasoning for these different settings.

The rationale for selecting the trigger of global cumulative deaths was two-fold – firstly, that because many countries implemented NPIs in response to COVID-19 deaths elsewhere in the world, globally reported deaths is a more appropriate metric than nationally reported deaths in triggering BPSV utilisation; and secondly that each accumulated death gives us more information about the severity of the threat being faced (and thus whether a response involving BPSV stockpile activation is appropriate). We selected the threshold presented in the main text arbitrarily however, which is why we also included in the Supplementary Information a suite of sensitivity analyses that vary this assumed threshold and explore how it shapes BPSV impact.

With regards to the reviewer’s second question as to why pathogen detection was chosen for the hypothetical SARS-X scenario, this is because the SARS-X model follows transmission within a single country. In this context a global-death trigger (appropriate for the retrospective COVID-19 analysis) has no analogue. Instead, the BPSV is activated when hospitalisations reach 5 per day, an objective, locally observable marker that couples first laboratory detection with clear evidence the outbreak is gaining severity. This threshold is the earliest actionable cue a national authority can reliably monitor. Because the global spread and lethality of a hypothetical virus cannot be parameterised ex-ante, any worldwide-death counter would be purely speculative. A local hospitalisation trigger therefore avoids unverifiable assumptions while letting us test the upper bound of timely BPSV deployment.

When discussing the results of containment of SARS-1-like and SARS-2-like viruses, the role of asymptomatic transmission should be made more apparent.

We have added text to the Results section noting our assumptions of the degree of both asymptomatic and pre-symptomatic transmission associated with each of the virus archetypes.

Results: “We considered two “archetype” sarbecoviruses – one similar to SARS-CoV-1 (Fig 1B, mean generation time 12 days, 0% presymptomatic transmission, 0% asymptomatic infections) and one similar to SARS-CoV-2 (Fig 1C, mean generation time 6.75 days, 35% presymptomatic transmission, 15% asymptomatic infections).”

The authors write “to simulate different vaccination strategies focussed on outbreak containment (defined as a final outbreak size of <10,000 infected individuals).” Where this threshold comes from?

This threshold was arbitrarily selected to give us sufficient epidemiological resolution to distinguish between outbreaks where the vaccination strategy has brought the reproduction number to a value just below 1 (in which case, the outbreak might continue on for some time before going extinct) compared to a value just above 1 (in which case the outbreak will continue until sufficient population-level immunity has been acquired). We note that in practice, the vast majority of outbreaks simulated (seeded with single-digit number of infections) are controlled to final sizes of well below 10,000 infected individuals.

There are some typos in the manuscript:

1) On the line 197-198, the figure labels seem wrong. For Italy, Iran, and Bangladesh, the labels should be Fig 3E, 3F, 3G respectively instead of 3D, 3E, 3F. Please check the labels in the whole paragraph.

Addressed, thank you for flagging.

2) On the line 198, there should be a space between 'from' and '124,500'.

Addressed, thank you for flagging.

3) On the line 207, '...also varying the time the time...', 'the time' repeats twice.

Addressed, thank you for flagging.

Reviewer #1 (Remarks on code availability):

The codes are provided in R. As I don't really know R I cannot judge the code provided. Nevertheless, the github page is well organized.

Thank you.

Reviewer #2 (Remarks to the Author):

Reviewer #3 (Remarks to the Author):

The paper presents a modelling study to investigate the potential impact of stockpiled broadly protective vaccine (with reduced efficacy comparatively to a targeted vaccine) in reducing disease burden in the event of a future SARS-X outbreak. The paper is well presented, and with its focus on pandemic preparedness, timely.

We appreciate the kind words, thank you. And more generally, thanks for all your feedback! Incorporating the edits and changes you suggested (described below) has, we feel, materially improved the manuscript – thank you for taking the time to review and for your constructive, useful suggestions.

The first sections, concerned with using a branching process model to assess the potential for viral containment, were the weakest both in terms of approach and relevance of the results. As a containment strategy, ring vaccination seems to lack plausibility. During the covid-19 pandemic, where early cases were detected and contacts identified, containment was attempted universally through the use of isolation (which is theoretically 100% efficacious against infection). Given that this was still only partially successful, the idea of replacing isolation of contacts in the early stages of a pandemic with a low efficacy vaccine seems unrealistic.

*Thank you for flagging these concerns – we agree that the presentation of ring-vaccination as a standalone strategy in the initial version of the manuscript lacked plausibility. In response to these concerns, we have completely overhauled its implementation to explicitly include representation of quarantine and isolation, parameterised using work carried out during the SARS-CoV-2 pandemic ([https://www.thelancet.com/article/S1473-3099\(20\)30457-6/fulltext](https://www.thelancet.com/article/S1473-3099(20)30457-6/fulltext)). In the revised version of the manuscript, we present an updated series of analyses (in the new versions of **Figures 1 and 2**, as well as **Supplementary Figures S1-S8**) exploring the impact of the BPSV on its own and in tandem with isolation and quarantine on outbreak containment prospects, for a suite of different assumptions about the efficacy of isolation and quarantine at preventing onwards transmission.*

Combining existing isolation strategies with spatially targeted BPSV in the early stages of an outbreak may have significant value however. The results presented demonstrate that both spatially targeted and ring vaccination are unlikely to contain an epidemic, even when it results from an identified index case, however it would be valuable to know how effective a spatially targeted BPSV campaign would be in slowing an early outbreak (effect on R_0 ?). BPSV use in high risk areas for importation (or in the locality of identified early cases) may have significant benefits in slowing down early outbreaks, increasing the chances of containment through case/contact isolation, and providing more time to develop other counter measures.

*Agreed! We have updated presented results associated with both ring and spatially targeted vaccination strategies to better quantify and describe their impact on epidemic dynamics. This includes quantification of the average reduction in the reproduction number (in the new versions of **Figures 1 and 2**, as well as **Supplementary Figures S1-S8**), as well as changes to the speed of the epidemic (**Supplementary Figures S1-S8**), which the reviewer well notes might have benefits even if the outbreak is not brought*

completely under control. Thanks to the reviewer for highlighting this – we feel these additions add an important element of nuance to the results beyond simply whether an outbreak is contained or not.

My suggestion would be to reimplement the branching process model to reflect this application; lose the focus on ring vaccination and instead show how effective spatially targeted BPSV may be in reducing R_0 in the early stages of an outbreak, and discuss the benefits this may bring.

We appreciate the constructive and thoughtful suggestion from the reviewer here – however we believe there is value to the inclusion of the ring-vaccination framework and results given the additions we have now made to the framework that align it better with real-world implementations of this strategy. Between these modifications and changes to both the analyses carried out and how the results are presented (including new figures describing reductions in the reproduction number and epidemic speed associated with the different strategies), we now believe the results to be of sufficient worth to be included.

If left in their current presentation, results from the branching process model also seem to be misleading in presenting a comparison between two different vaccination strategies. Each strategy relies on a completely different set of assumptions and tested for different sensitivities. For instance the spatial strategy is triggered upon a hospitalisation threshold (meaning a potentially large number of initial cases) and ring vaccination is more ambitiously triggered upon a symptomatic index case.

*We completely agree with the reviewer that presenting both ring and spatially targeted vaccination strategies side-by-side is misleading given they are two completely different strategies underpinned by two different set of assumptions. Given this, as well as the expansion of analyses carried out for each strategy, we have made the decision to split the previous **Figure 1** of the manuscript into two separate figures, each detailing the analyses and results for a single vaccination strategy. We hope that this strengthens the results associated with each and sufficiently differentiates them from each other.*

Finally, where the branching process model is used, it would be good to see some discussion of the limitations of the model. It may be an effective simplification in the early stages of an outbreak, but given the weight of evidence to support the importance of household infections within the covid pandemic, the branching model is likely insufficient to capture the true network structure and it is important to understand how this may affect the results presented.

The reviewer raises an important point – we have added text in both the Results and Discussion sections of the manuscript highlighting these limitations (and others associated with the branching process framework) in detail.

Discussion: “An important caveat to these results is the simplifying assumption that omits the strong clustering of infections within households (where secondary-attack rates for SARS-CoV-2 were several-fold higher than in community settings⁴¹) and the complex network structure over which pathogens spread through populations. Ignoring this structure may slightly underestimate vaccine reach (household contacts are easier to identify and vaccinate) but overestimate the additional benefit that quarantine confers (given that isolation is typically less effective at reducing within household transmission compared to transmission outside the household).”

The remainder of the results were based on a compartmental covid model fitted to 185 countries. This model has already had substantial use elsewhere. The results from this model were generally compelling and well presented.

Thank you!

My main criticism here would be the assumption that all countries manage an equally fast BPSV deployment. While I strongly support the importance of showing equity in global vaccine coverage and availability, we are a long way off being able to have the infrastructure to deploy vaccines equally fast across the globe; this is due to challenges not just in resources, but also infrastructure, population structure, and environmental factors. The reality then could significantly impact results, and this would ideally be taken into account in the model.

Thanks for raising this important point. An important point that was perhaps not clear in the initial manuscript was that we actually do incorporate different vaccination rates across countries (specifically, we use World Bank income-strata specific averages for each LIC, LMIC, UMIC and HIC). We have amended the Methods text to highlight this. Additionally, we have added a further sensitivity analysis for the COVID-19 retrospective impact analysis where vaccine deployment rates are kept constant across all countries, so that they can be compared to the scenario the reviewer suggests (and which the manuscript already contained).

Finally, while I understand the desire for brevity and avoiding repeating details shown elsewhere. I would've found more model details helpful throughout, especially in the supplement. In particular detailing all the parameters in the model, where they come from, how they're fitted, which ones differ between countries, where does uncertainty come from in the model?

Addressed - in response to this feedback, as well as the feedback from Reviewers 1 and 2, we have added significant further information on model details in the Supplementary Information, especially to the descriptions of the branching process model and ring/spatially targeted vaccination strategies, given that work has not been published previously. This clarifies the model details (particularly the formulation and parameterisation) in more detail.

A Fig 5 is referred to in the supplement, but does not exist, can you check this label.

Addressed, thank you for flagging.

In Fig 2c (and similar) can you put the crosses in front on the dots, so it is obvious whether they are actually the same or just absent on some points?

Addressed, thank you for flagging.

REVIEWER COMMENTS

REVIEWERS' COMMENTS

Reviewer #1 (Remarks to the Author):

The author did a good job revising the manuscript and addressing most points raised in the first round of review. A few points however need still attention:

In the revised version, explicit mathematical equations describing the epidemic dynamics are not included. The authors opted to cite a reference for the model. However, upon review, that reference also lacks mathematical equations and instead points to another source. Despite being published somewhere, it would be important to add these details of the model in this paper (of course in the SI) to make the publication self-contained.

Addressed - in response to this feedback, we have enclosed the full mathematical equations for the model in the Supplementary Information.

Regarding the uncertainty, the authors state that data fitting was only performed for the retrospective SARS-CoV-2 in Figure 4. Nevertheless, confidence intervals should also be provided for the low, moderate, and high coverage scenarios in Figure 4 panels (E), (F), and (G), in addition to the original no-BPSV case.

We appreciate the reviewers' point here but it is not possible to plot the confidence intervals for the coverage mortality trajectories without significant overlap that we believe obscures and compromises the readability of the graph. We have however noted in the Figure legend that what we have plotted in the mean trajectory for that particular coverage scenario; we also note that the uncertainty in these trajectories across all countries is captured in the estimates and corresponding confidence intervals plotted in Figure 4C.

In figure 4, the description for panel (C) is missing, and the labels currently designated as (C), (D), and (E) should be shifted to correspond to panels (D), (E), and (F), respectively.

Thanks for flagging this oversight – we've corrected this mistake and added in the description for panel (C).

Reviewer #1 (Remarks on code availability):

I don't really know R hence I cannot comment on the details of the code. The github looks well organized though

Thanks! We have also created a DOI via Zenodo so users can access the exact version of the repo used to produce the analyses.

Reviewer #2 (Remarks to the Author):

Reviewer #2 (Remarks on code availability):

The authors have provided a clear README file with instructions for installation and execution in the repository.

Thanks! As above, note that in addition to functional organisation of the repository, we have also created a DOI via Zenodo so users can access the exact version of the repo used to produce the analyses.

Reviewer #3 (Remarks to the Author):

My thanks to the authors for taking time to consider my original feedback, they have done a great job of addressing my concerns and suggestions. The revised manuscript appears much more robust, relevant, and applicable. I only have a few very minor new comments.

The thanks are entirely ours for the incredibly helpful comments that materially improved the manuscript!

In figure's 1 and 2, can sensitivity to lower vaccine efficacy also be shown? I.e. Given an assumed vaccine efficacy of 35% it would seem more sensible to present the heatmaps in fig.1/2G to span between 0-70% unless there is good reason that higher efficacy vaccines are likely.

Also, in these heatmaps, I assume the reference scenario looks different between scenarios due to stochasticity? (could the same simulation set be used for each of these?). How many simulations are used for these. In all the results it would be beneficial to present the effects of stochasticity more (prediction intervals shown on lines/bars when it doesn't negatively impact readability). The confidence intervals in 4E-F seem very narrow, though this may well be due to model limitations (dynamics that are not considered).

Re figure 1 and 2: the choice of 35% was informed by conversations with a number of different teams at CEPI involved in supporting development of broadly protective sarbecovirus candidates. We agree exploring a wider range would be helpful, and so have extended the heatmaps to go down to 15% - 75% (with the upper value representing the assumed efficacy against severe disease used in other analyses and which forms a clear upper bound given that efficacy against infection must be lower than efficacy against severe disease). These updated heatmaps are now included in the new figures.

Re the heatmaps: Thanks for flagging this – we have now clarified in the figure legend for the heatmaps that the results plotted are the proportion of outbreaks contained from 100 stochastic simulations. As these are proportions (and thus precision scales arbitrarily with number of simulations), there

Wherever we plot a quantitative outcome which varies across stochastic simulations (e.g. Ref), we note that we do include error bars to indicate and highlight to the reader the extent of stochasticity and variation across simulations. We have also added text to the figure legend clarifying this further, and note the inclusion of extensive sensitivity analyses in the Supplementary which further contextualise the sources and drivers of variation in the results presented.

Re Figure 4E-4F: you are correct in that the confidence intervals are narrow. This indeed reflects model limitations (namely that in fitting to 180+ countries, only variation in the R_t trajectory is considered).

It is now stated in the limitations that using a branching process model, rather than the true network structure may underestimate vaccine reach since “household contacts are easier to identify and vaccinate”. Perhaps more significantly, it should be said that a branching model underestimates vaccination impact since it overestimates the size of the exposed population (generation n will feature many of the same individuals already seen in generations 1 to $n-1$).

We have added text in noting this, thanks!

“This is because household contacts are easier to identify and vaccinate, and additionally, because a branching process without household structure fails to account for the substantial overlap of contacts across successive generations of infections belonging to the same household. Consequently, any fixed stockpile of vaccine doses will cover a larger proportion of the actual at-risk population, increasing the effectiveness of such vaccination strategies.”